# Stat3 promotes mitochondrial transcription and oxidative respiration during maintenance and induction of naïve pluripotency

Elena Carbognin[1,†], Riccardo M Betto[1,†], Maria E Soriano[2], Austin G Smith[3,4,*] & Graziano Martello[1,**]

## Abstract

Transcription factor Stat3 directs self-renewal of pluripotent mouse embryonic stem (ES) cells downstream of the cytokine leukemia inhibitory factor (LIF). Stat3 upregulates pivotal transcription factors in the ES cell gene regulatory network to sustain naïve identity. Stat3 also contributes to the rapid proliferation of ES cells. Here, we show that Stat3 increases the expression of mitochondrial-encoded transcripts and enhances oxidative metabolism. Chromatin immunoprecipitation reveals that Stat3 binds to the mitochondrial genome, consistent with direct transcriptional regulation. An engineered form of Stat3 that localizes predominantly to mitochondria is sufficient to support enhanced proliferation of ES cells, but not to maintain their undifferentiated phenotype. Furthermore, during reprogramming from primed to naïve states of pluripotency, Stat3 similarly upregulates mitochondrial transcripts and facilitates metabolic resetting. These findings suggest that the potent stimulation of naïve pluripotency by LIF/Stat3 is attributable to parallel and synergistic induction of both mitochondrial respiration and nuclear transcription factors.

**Keywords** LIF; metabolism; mitochondrial respiration; pluripotency; Stat3
**Subject Categories** Metabolism; Stem Cells
**The EMBO Journal (2016) 35: 618–634**

## Introduction

Mouse embryonic stem (ES) cells (Evans & Kaufman, 1981; Martin, 1981) have the capacity to give rise to all differentiated cells of the body and the germ line (Bradley *et al*, 1984), a feature termed pluripotency (Bradley *et al*, 1984; Martello & Smith, 2014). ES cells are derived from the naïve pluripotent epiblast of mouse blastocysts (Brook & Gardner, 1997; Boroviak *et al*, 2014). The cytokine leukemia inhibitory factor (LIF) is pivotal for establishing and maintaining ES cells in culture (Smith *et al*, 1988; Williams *et al*, 1988; Nichols *et al*, 1994). LIF signals via the LIF-R/gp130 complex, which activates Janus-associated kinases (JAKs) (Burdon *et al*, 2002). In turn, JAKs phosphorylate and activate the transcription factor Stat3, which maintains naïve pluripotency through its direct targets Tfcp2l1, Klf4, and Gbx2 (Niwa *et al*, 1998, 2009; Bourillot *et al*, 2009; Martello *et al*, 2013; Tai & Ying, 2013), key members of the ES cell core gene regulatory network (Dunn *et al*, 2014).

Blockade of GSK3 and MEK kinases permits ES cell self-renewal in the absence of LIF (Ying *et al*, 2008; Martello *et al*, 2013). Importantly, however, self-renewal efficiency is significantly increased when LIF is added (Wray *et al*, 2010; Dunn *et al*, 2014). LIF/Stat3 signaling is also critical during cellular reprogramming to facilitate the attainment of naïve pluripotency (Takahashi & Yamanaka, 2006; Yang *et al*, 2010; van Oosten *et al*, 2012; Martello *et al*, 2013; Stuart *et al*, 2014).

Naïve pluripotent cells are metabolically flexible as they utilize both glycolysis and mitochondrial respiration (Zhou *et al*, 2012; Teslaa & Teitell, 2015). However, this is not a feature of all pluripotent cells. EpiSCs derived from the primed epiblast of post-implantation embryos (Brons *et al*, 2007; Tesar *et al*, 2007; Nichols & Smith, 2009) are mainly glycolytic with inert mitochondria (Zhou *et al*, 2012). The switch from aerobic to anaerobic metabolism presumably reflects the altered environment of the embryo upon implantation, but is evidently intrinsically programmed.

Here, we investigate the impact of LIF/Stat3 on mitochondrial activity during mouse ES cell propagation and reprogramming from primed to naïve pluripotency.

## Results

### The LIF/Stat3 axis promotes ES cell proliferation and mitochondrial transcription

Embryonic stem cells can be derived and expanded under feeder-free conditions in the presence of two inhibitors (2i) with or without

1 Department of Molecular Medicine, University of Padua, Padua, Italy
2 Department of Biology, University of Padua, Padua, Italy
3 Wellcome Trust – Medical Research Council Cambridge Stem Cell Institute, University of Cambridge, Cambridge, UK
4 Department of Biochemistry, University of Cambridge, Cambridge, UK
  *Corresponding author. Tel: +44 1223 760233; E-mail: austin.smith@cscr.cam.ac.uk
  **Corresponding author. Tel: +39 49 8276088; E-mail: graziano.martello@unipd.it
  †These authors contributed equally to this work

LIF (Ying *et al*, 2008; Wray *et al*, 2010). ES cells expanded in 2i retain the ability to form chimeric animals and be transmitted through the germline, indicating that LIF signaling can be dispensable for the maintenance of pluripotency *in vitro*, although LIF dependency varies with genetic background. Consistent with this, *Stat3* null ES cells have been previously derived and characterized in 2i and showed no overt defects in early lineage differentiation or self-renewal capacity (Ying *et al*, 2008; Wray *et al*, 2011; Martello *et al*, 2013). Nonetheless, addition of LIF to 2i (2i + LIF) is beneficial to the culture of wild-type ES cells, resulting in increased clonogenicity (Wray *et al*, 2010; Dunn *et al*, 2014) and more robust and rapid expansion (Fig 1A).

We investigated whether the effect of LIF on population doubling was due to an increase in cell survival or in proliferation rate. We found that the percentage of viable cells was not affected (Appendix Fig S1A), but that LIF caused a reduction in the fraction of cells in G1 phase, with a concomitant increase in actively dividing cells (Fig 1B).

LIF is known to activate three signaling pathways, Stat3, PI3K, and Erk, each of which could mediate an effect on proliferation (Burdon *et al*, 1999). Presence of the Mek inhibitor in 2i rules out a contribution of the Erk cascade. We took advantage of *Stat3* null cells and found that their proliferation rate is not increased by LIF and is comparable to that of wild-type cells cultured without LIF (Fig 1A). We conclude that Stat3 is required for the proliferative response to LIF.

We analyzed transcriptome data from mES cells cultured in 2i and stimulated with LIF for 1 h (Martello *et al*, 2013) to identify transcriptional targets that might be related to the effects on proliferation. We found that several mitochondrial transcripts were elevated in response to LIF (Fig 1C). In particular, mRNAs coding for subunits of the complexes of the mitochondrial respiratory chain were upregulated around twofold by LIF treatment (Fig 1D). This effect was not observed in *Stat3* null cells. These results were validated by quantitative real-time PCR (RT–qPCR) on cells either acutely stimulated with LIF or kept in 2i + LIF conditions for 2 passages, the latter result indicating that the response is stable over time (Fig 1E, top).

LIF/Stat3 could enhance mitochondrial transcription indirectly, via induction of known mitochondrial master transcriptional regulators, such as PGC-1 or TFAM. Inspection of the RNA-seq data from LIF stimulation showed no induction of either of these regulators (Appendix Fig S1C).

To explore whether the effect of LIF/Stat3 on mitochondrial transcription may be direct, we designed a reporter assay. A single regulatory region, the D-loop, directs transcription of the mitochondrial genome. We generated a reporter construct containing the mouse D-loop followed by a minimal promoter and the firefly luciferase ORF (D-loop-Lux, Fig 2A) and introduced this into both ES cells and EpiSCs. In either case, cotransfection with Stat3 increased reporter activity (Fig 2B and C). EpiSCs showed more pronounced reporter activation, probably due to lower levels of endogenous Stat3 pathway.

To examine further whether Stat3 could directly regulate mitochondrial transcription, we inspected available chromatin immunoprecipitation followed by sequencing (ChIP-seq) data (Sánchez Castillo *et al*, 2015). We observed a significant enrichment of Stat3 over the D-loop region of the mitochondrial genome (Fig 2D). We performed ChIP-qPCR and confirmed binding of Stat3 at the D-loop in mES cells (Fig 2E).

Mitochondrial genomes exist as clusters associated with specific proteins, termed nucleoids, that lie within the mitochondrial matrix. Atad3 is a protein required for correct nucleoid assembly which interacts with the D-loop region (He *et al*, 2007). We first confirmed that Atad3 and mtDNA colocalized in mES cells (Appendix Fig S2A). We used the proximity ligation assay (PLA) to test for colocalization of endogenous Stat3 and Atad3. The results in Fig 2F indicate that the two proteins are closely associated within mitochondria in ES cells.

Collectively, these findings suggest that Stat3 directly induces transcription of the mitochondrial genome, but do not rule out other potential effects of Stat3 on the stability or turnover of mitochondrial transcripts.

### Mitochondrial respiration is increased in the presence of LIF

We investigated whether alterations in the level of mitochondrial transcription are accompanied by altered respiratory activity. Stat3 was previously shown to be a positive regulator of mitochondrial respiration in terminally differentiated cells (Wegrzyn *et al*, 2009; Zouein *et al*, 2014). We measured the oxygen consumption rate (OCR) in wild-type and *Stat3* null cells cultured in 2i + LIF by extracellular flux analysis (Seahorse assay). In the absence of Stat3, we found a reduction both in the basal levels of OCR and after treatment with the uncoupler FCCP, which provides a measure of the maximal respiratory rate (Figs 3A and Appendix Fig S3A). These

**Figure 1.  LIF/STAT3 signaling promotes proliferation and mitochondrial transcription.**

A   Proliferation assay of Stat3$^{+/+}$ and Stat3$^{-/-}$ cells cultured in N2B27-based 2i media either in the presence or in the absence of LIF. Cells were seeded and scored for four consecutive days. Scores were normalized to day 1. Mean and s.d. of three technical replicates. See also Appendix Fig S1A.

B   Cell cycle analysis of Stat3$^{+/+}$ cells cultured in 2i media without LIF or with LIF for several passages and Stat3$^{-/-}$ cells in 2i with LIF. Forty-eight hours after plating, cells were detached, treated with propidium iodide, and analyzed by flow cytometry. Top: Cells cultured in the presence of LIF showed an increment in S/G2 and a decrease in G1. Mean and s.e.m. of three technical replicates. Unpaired *t*-test: *$P < 0.05$. Bottom: representative plots.

C   Scatter plot showing RNA-seq data from Stat3$^{+/+}$ cells cultured in 2i and stimulated with LIF for 1 h (Martello *et al*, 2013). Absolute expression is shown in RPKM. Green dots indicate known LIF targets that serve as positive controls. Mitochondrial-encoded transcripts are represented as orange dots. Only genes with FC > 1.7 and a *P*-value < 0.05 are shown.

D   Heatmap showing mean normalized expression of 13 mitochondrial transcripts encoding 4 subunits of the mitochondrial respiratory chain. RNA-seq data are from Stat3$^{+/+}$ and Stat3$^{-/-}$ cells were expanded in 2i media without LIF and treated with LIF for 1 or 24 h.

E   Gene expression analysis by RT–qPCR of Stat3$^{+/+}$ (blue) and Stat3$^{-/-}$ (red) cells cultured in 2i and treated with LIF for 1 h, 4 h or 4 days. Data are normalized to unstimulated 2i cultures. Beta-actin served as an internal control. Mean and s.e.m. of three independent experiments. Unpaired *t*-test: *$P < 0.05$, **$P < 0.01$, ***$P < 0.001$. See also Appendix Fig S1B.

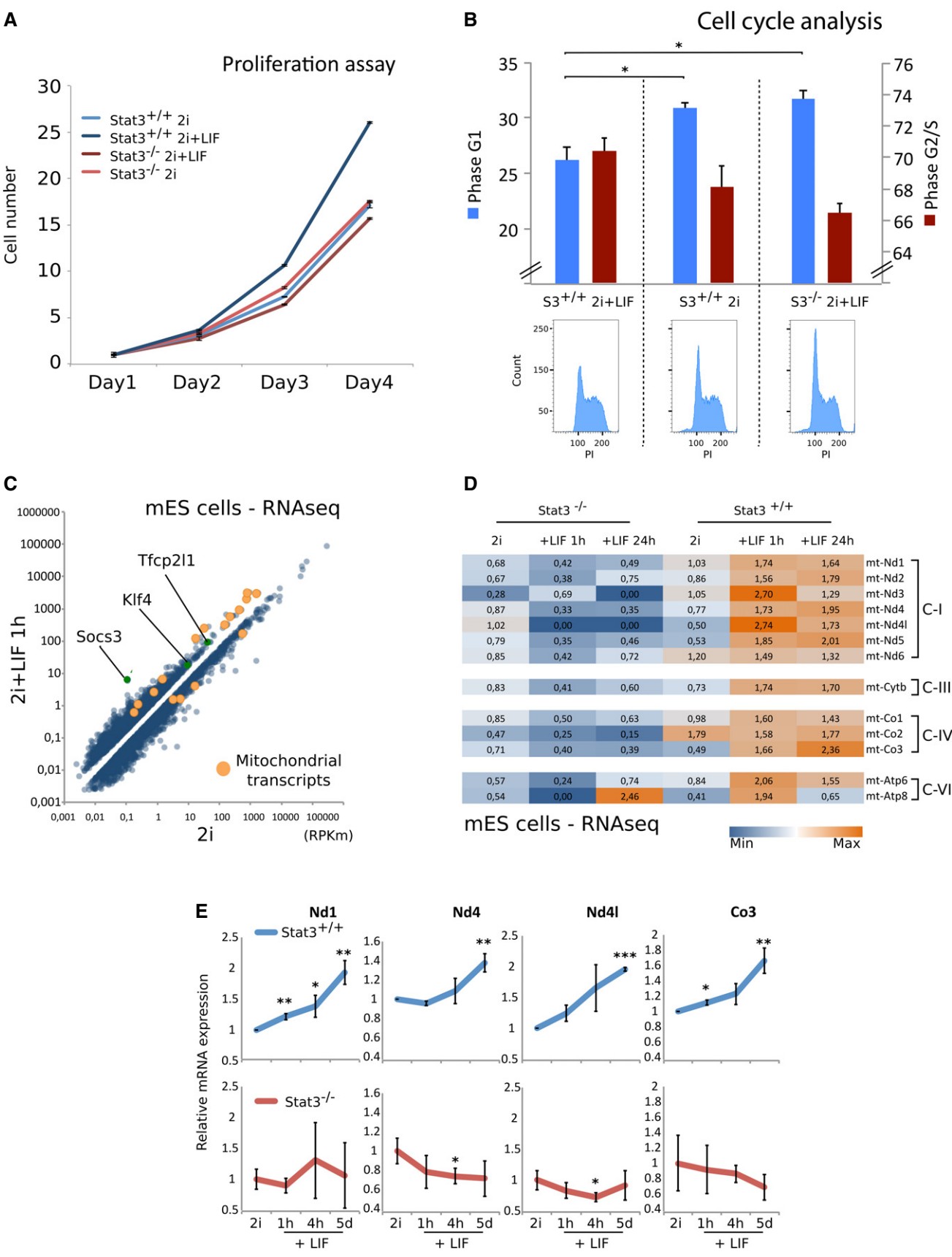

**Figure 1.**

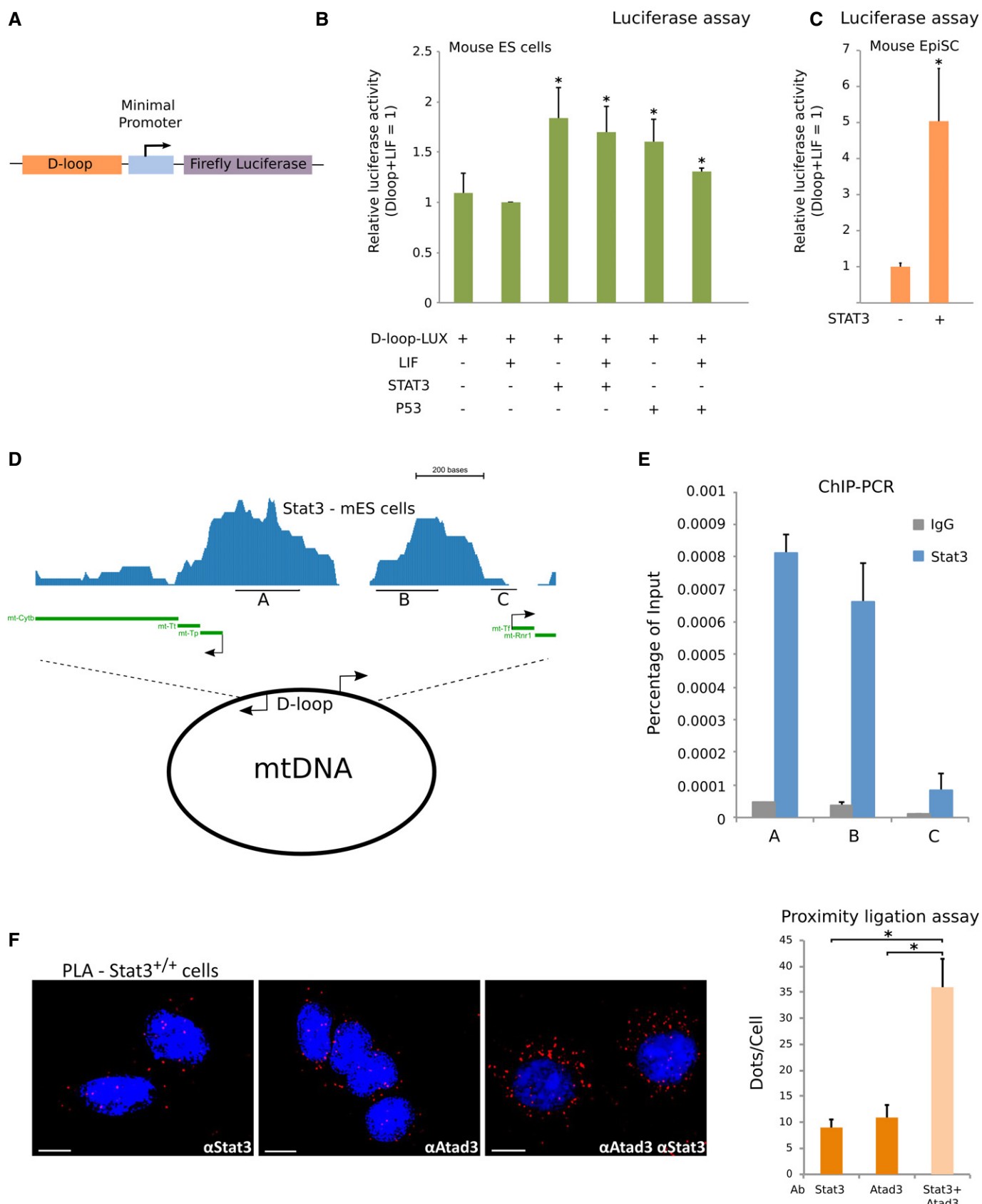

**Figure 2.**

◀

**Figure 2.   Stat3 regulates directly the mitochondrial DNA.**

A   Schematic representation of D-loop-Lux reporter construct used for luciferase assays.

B   Luciferase assay on ES cells transfected with D-loop-Lux reporter plasmid and Stat3 in the presence or in the absence of LIF for 48 h; p53 was previously shown to activate a similar reporter construct (Heyne *et al*, 2004) and therefore was used as a positive control. Increased expression of Stat3 enhances luciferase activity. Mean and s.e.m. of four independent experiments. Unpaired *t*-test: *$P < 0.05$.

C   Luciferase assay on EpiSCs transfected with D-loop-Lux reporter plasmid and Stat3 in the presence of LIF for 48 h. Mean and s.e.m. of three independent experiments. Unpaired *t*-test: *$P < 0.05$.

D   Diagram highlighting available ChIP-seq results (GSM288353) of Stat3 binding on the D-loop. A, B, and C indicate regions where primers for ChIP-PCR analysis were designed.

E   Chromatin immunoprecipitation (ChIP) was performed using anti-Stat3 or a rabbit control IgG antibody in Rex1-GFP cells (Wray *et al*, 2010) cultured in LIF and serum conditions in the presence of blasticidin to reduce heterogeneity of the culture. ChIP-PCR was performed with primers located on three regions of the D-loop (A, B, C), as indicated in (D). Mean and s.d. of two independent experiments are shown.

F   Representative confocal images of Stat3$^{+/+}$ cells subjected to proximity ligation assay (PLA) by using anti-Stat3 and anti-Atad3 antibodies (see Appendix Fig S2 for Atad3 localization in nucleoids). Only when the two proteins are close to each other, an enzymatic reaction takes place, producing discrete fluorescent red dots in the nanometer range (bottom). Anti-Stat3 or anti-Atad3 alone was used to assess the assay specificity (top and center, respectively). DAPI serves as a nuclear counterstain. Scale bar, 10 μm. The histogram shows the quantification of PLA performed on Stat3$^{+/+}$ cells. The number of red dots/cell is plotted. Note that when cells are stained with both antibodies, the number of dots increases significantly, suggesting close proximity between Stat3 and the nucleoids marker Atad3 (light orange bar). Mean and s.e.m. of > 15 cells for each sample are shown. Unpaired *t*-test: *$P < 0.05$.

results prompted us to assess whether the positive effect of Stat3 on mitochondrial respiration requires active LIF signaling or may be a constitutive function of Stat3 independent of the signaling context. We measured OCR in cells cultured for multiple passages in either 2i or 2i + LIF and observed an increase in both basal and maximal respiration in the presence of LIF (Fig 3B and C). Under the same conditions, we measured the extracellular acidification rate (ECAR), which provides an indirect measure of the glycolytic flux, and found that LIF has no consistent effect on ECAR (Appendix Fig S3B and C).

Increased respiration could be due to enhanced mitochondrial biogenesis. However, protein levels of two components of the import machinery (TOM20 and TIMM23), whose expression correlates with mitochondrial biomass, were not increased in the presence of LIF (Fig 3D), suggesting that LIF does not have a substantial influence on mitochondrial biogenesis. We also measured the number of copies of the mitochondrial genome relative to the nuclear genome by PCR in 2i or 2i + LIF and could not detect any significant difference (Fig 3E). A constant number of genomes are consistent with the elevated mitochondrial transcript levels arising from a specific increase in transcription. We focussed our attention on Complex I, which is the main entry point to the respiratory chain, because several of its subunits are transcriptionally regulated by LIF/Stat3 (Fig 1C and E). We performed blue native gel electrophoresis (BNGE) to isolate intact complexes and by Western blot observed a reduction in the levels of Complex I in the absence of LIF and still lower levels in *Stat3* null cells (Fig 3F). BNGE also allows the detection of high molecular weight clusters containing several complexes, called supercomplexes (Schägger, 1995). We observed that supercomplexes are present in ES cells and that their levels are reduced in the absence of either LIF or Stat3 (RCS on Fig 3F and G). These results suggest that LIF/Stat3 increases the levels of complexes of the respiratory chain, which in turn results in enhanced assembly of supercomplexes and elevated mitochondrial respiration.

## Mitochondrial respiration determines optimal proliferation

LIF/Stat3 promotes the proliferation of ES cells as well as mitochondrial respiration. We asked whether the two effects are causally linked. To this end, we first applied rotenone, an inhibitor of Complex I. We titrated rotenone and found that concentrations ranging from 50 to 100 nM were able to reduce cell proliferation (Appendix Fig S4A), also reducing OCR by ~70% without affecting ES cell viability (Appendix Fig S4B and Fig 4A). We then tested the effect of rotenone on proliferation upon perturbation of the LIF/Stat3 axis (Fig 4B). LIF increased the number of wild-type cells (Fig 4B, compare 1st and 2nd bar) and rotenone abrogated this effect (compare the 2nd bar to the 3rd and 4th bars). As expected, *Stat3* null cells did not respond to LIF (5th vs. 6th bar), but they also appeared more affected by rotenone (compare 6th to the 7th and 8th bars), a result in line with their reduced basal respiratory capacity (Fig 3A). Similar results were obtained with wild-type ES cells cultured in LIF + serum (Appendix Fig S4C), suggesting that the effect described is not related to the 2i culture conditions.

As an independent test, we depleted Ndufs3, a Complex I subunit that has been shown to be required for Complex I assembly and activity (Lapuente-Brun *et al*, 2013), using shRNA. Ndufs3 knockdown resulted in reduced OCR levels and proliferation in response to LIF (Fig 4C–E and Appendix Fig S4D), consistent with the rotenone results.

To further confirm that the effects of rotenone are due to inhibition of the respiratory chain, we used antimycin A, an inhibitor of Complex III. Titrated doses of antimycin A were sufficient to reduce OCR with no effect on cell survival (Appendix Fig S4E–G) and also potently reduced cell proliferation (Fig 4F). Similar results were obtained with myxothiazol, a second Complex III inhibitor (Appendix Fig S5A–C).

Inhibition of the respiratory chain could affect the production of reactive oxygen species (ROS), which in turn could either be cytotoxic or act as signaling molecules. We assayed the production of ROS after treatment with rotenone, antimycin A, and myxothiazol and did not detect any increase (Appendix Fig S6A–D) at the concentrations that affected ES cell proliferation. Therefore, ROS does not seem to play a role in this context.

We then tested the effects of long-term treatment with rotenone. We observed a dose-dependent reduction in the cumulative number of cells over multiple passages (Fig 4G), but without overt effects on cell survival (Appendix Fig S6E). Crucially, ES cells remained morphologically undifferentiated and maintained full expression of pluripotency factors (Fig 4H and I). Moreover, known direct transcriptional targets of LIF/Stat3 were not affected, suggesting that

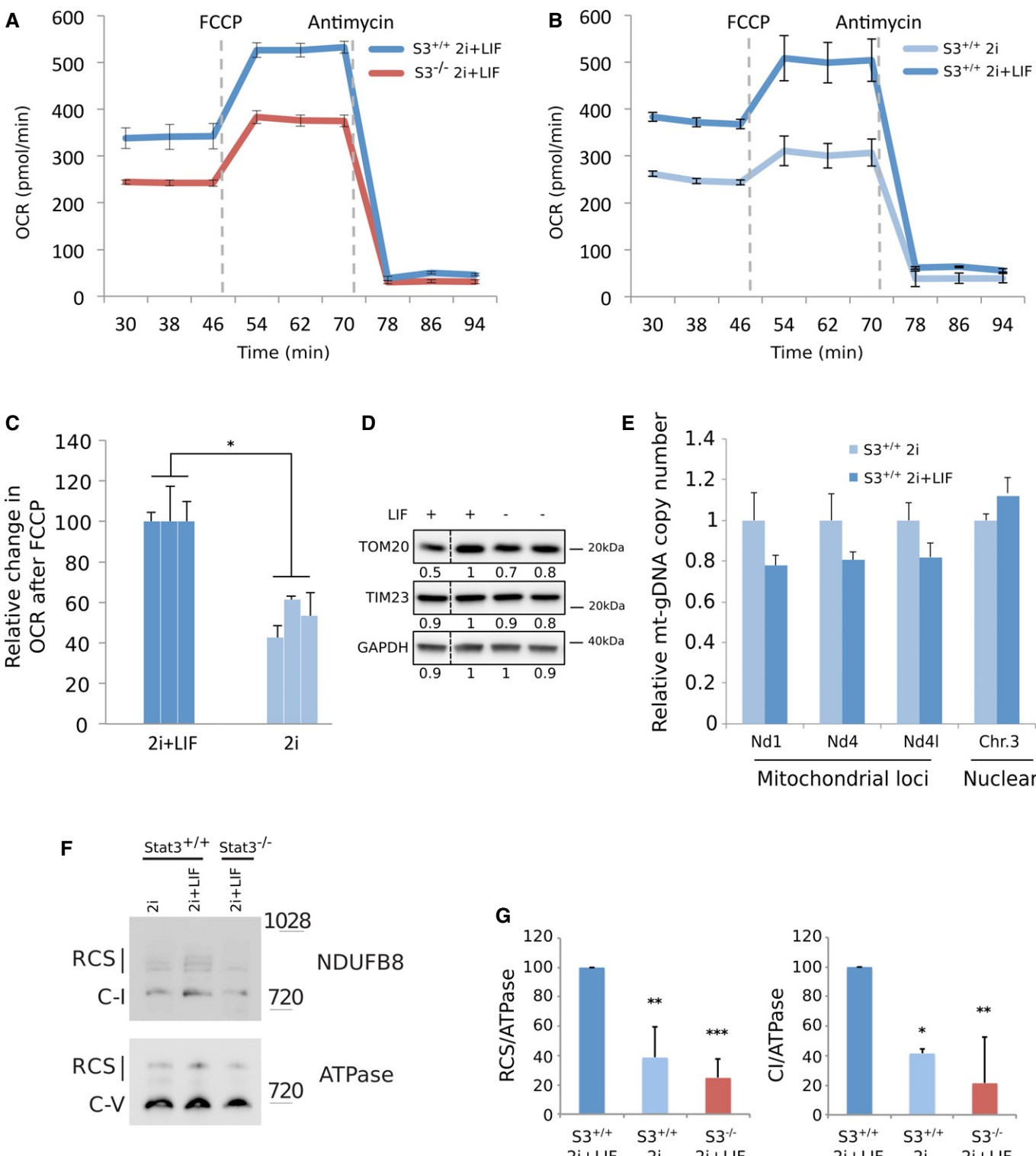

**Figure 3.**

rotenone does not affect LIF signaling to the nucleus in ES cells (Fig 4H, right bars).

These results indicate that mitochondrial respiration is instrumental for maximal proliferation of ES cells and furthermore suggest that LIF effects on proliferation and pluripotent cell identity may be uncoupled.

**Mitochondrial localization of Stat3 is crucial for LIF effects on proliferation**

The effects of LIF signaling on ES cell proliferation and mitochondrial activity are strictly dependent on the presence of Stat3 (Figs 1A and B, and 3A). Thus, *Stat3* null cells represent a valuable tool. We

◀

**Figure 3. LIF/Stat3 activates mitochondrial respiration.**

A  Oxygen consumption rate (OCR) measured by Seahorse extracellular flux assay of Stat3$^{+/+}$ and Stat3$^{-/-}$ cells maintained in 2i condition in the presence of LIF; 200 nM FCCP (a mitochondria uncoupler) treatment resulted in higher OCR increase in Stat3$^{+/+}$ compared to Stat3$^{-/-}$ cells, showing a higher level of maximal mitochondrial electron transport chain (ETC) activity in Stat3$^{+/+}$ cells. Injection of 200 nM antimycin shows similar non-mitochondrial respiration rates for both Stat3$^{+/+}$ and Stat3$^{-/-}$ cells. Mean and s.e.m. of 5 technical replicates are shown.

B  Oxygen consumption rate (OCR) of Stat3$^{+/+}$ cells cultured in 2i conditions without LIF or with LIF for several passages; 200 nM FCCP and 200 nM antimycin were injected and resulted in a higher mitochondrial respiration activity in cells cultured in the presence of LIF. Mean and s.e.m. of 4 replicates are shown. See also Appendix Fig S3D.

C  Relative changes in oxygen consumption after 200 nM FCCP treatment of Stat3$^{+/+}$ cells cultured in 2i media in the presence (dark blue bars) and absence of LIF (light blue bars). Mean and s.e.m. of > 4 technical replicates of three independent experiments are shown. Unpaired *t*-test: *$P < 0.05$.

D  Western blot of Stat3$^{+/+}$ cells cultured in the presence or absence of LIF. Note that protein levels of two mitochondrial markers (TOM20 and TIMM23) do not change in the absence of LIF. GAPDH was used as a loading control. Relative mean intensity is shown below each band.

E  Mitochondrial DNA expression analysis of Stat3$^{+/+}$ cells maintained in 2i in the presence (dark blue bars) or absence (light blue bars) of LIF. The abundance of 3 mitochondrial genomic loci was measured and normalized to a nuclear genomic locus on chromosome 3. Mean and s.e.m. of three independent biological replicates are shown.

F  BNGE analysis followed by Western blot for a Complex I protein (NDUFB8). ATPase serves as a loading control. RCS, respiratory chain supercomplexes. See also Appendix Fig S3E and F.

G  Quantification of the chemiluminescent RCS/ATPase signal ratio (left) and Complex I/ATPase signal ratio (right). Mean and s.d. of three independent experiments. Unpaired *t*-test: *$P < 0.05$, **$P < 0.01$, ***$P < 0.001$.

Source data are available online for this figure.

first transfected *Stat3* null cells with a full-length *mStat3* ORF, randomly picked several clones, and selected two clones expressing Stat3 protein at ~twofold over endogenous wild-type levels (Fig 5A and B). Both clones reacquired the ability to respond to LIF measured by activation of the direct target Socs3 (Fig 5C) and in terms of cell proliferation (Fig 5D). When challenged with rotenone, the rescue clones proliferated more than null cells (Appendix Fig S7A). We conclude that the proliferative defects observed in *Stat3* null cells are reversible and specifically due to the lack of Stat3.

We tested whether in ES cells the effects of LIF on proliferation and respiration are due to Stat3 localization to the mitochondria, or are mediated by nuclear targets of Stat3. To do so, we transfected *Stat3* null cells with a construct expressing the Stat3 cDNA fused to a mitochondrial localization signal (MLS-Stat3) as previously described (Wegrzyn *et al*, 2009). We generated clones expressing MLS-Stat3 at similar levels to endogenous Stat3 in wild-type cells (Fig 5E, total fractions). We prepared the mitochondrial fraction from ES cells. Representation of the nuclear protein TRIM33 was reduced by > 90% (Fig 5E) compared to total cell extracts, while the mitochondrial protein TOM20 was readily detectable, indicating successful isolation of mitochondria. Endogenous Stat3 protein was detected in the mitochondrial fraction of wild-type cells and MLS-Stat3 was clearly enriched in the mitochondria of transfected cells.

Immunofluorescence staining also indicated that MLS-Stat3 was present in mitochondria (Fig 5F). Conversely, double immunostaining with TOM20, a protein present on the outer mitochondrial membrane, shows adjacent but non-overlapping localization (Appendix Fig S7C), suggesting that MLS-Stat3 is located within the mitochondrial matrix.

We characterized the transcriptional response in cells expressing MLS-Stat3 and found that they did not activate the nuclear target Socs3 in response to LIF (Fig 5G, blue bars). In contrast, mitochondrial targets were activated at levels comparable to, or higher than, control cells (Fig 5G, yellow bars). By ChIP-PCR, we found a significant enrichment of MLS-Stat3 on the D-loop region of the mitochondrial genome (Fig 5H). A direct interaction between MLS-Stat3 and nucleoid structures was also evidenced by PLA (Fig 5I and Appendix Fig S7D).

Finally, we again measured the expression of master regulators of mitochondria transcription, such as TFAM, and found no significant changes (Appendix Fig S7E). Collectively, these data are consistent with direct regulation of expression of mitochondrial genes.

We examined the functional impact of MLS-Stat3 on proliferation of *Stat3* null cells. All three MLS-Stat3 clones expanded more rapidly than the null cells, and two of the clones showed a similar increase in cell numbers to wild-type cells (Fig 5J). MLS-Stat3 clones also proliferated more in the presence of rotenone (Fig 5K). They showed typical compact morphology of undifferentiated ES cells and colony sizes appeared on average larger than null cells (Fig 5L). These experiments were performed in the presence of LIF. In 2i without LIF, MLS-Stat3 clones exhibited a similar expansion rate to *Stat3* null cells (Appendix Fig S8A), indicating that the effect of MLS-Stat3 on ES cell proliferation requires LIF stimulation.

A minor fraction of MLS-Stat3 becomes phosphorylated on Tyr705, the JAK target site (Appendix Fig S7B), but whether this is in the mitochondria is uncertain and the mechanism for such an effect is unknown. We also noted that LIF increased total Stat3 protein levels (Appendix Fig S8B). While Stat3 is known to autoregulate its own transcription, this will not apply to the MLS-Stat3 transgene. This observation therefore suggests that another mode of LIF signaling, potentially through PI3K, may increase translation of MLS-Stat3 or stabilize the protein.

We examined whether MLS-Stat3 is able to mediate the effects of LIF on inhibition of ES cell differentiation, which is considered to be dependent on nuclear transcriptional targets (Niwa *et al*, 2009; Martello *et al*, 2013). *Stat3* null cells and MLS-Stat3 clones were transferred to culture in LIF and Mek inhibitor (LIF + PD), conditions that are sufficient for wild-type ES cell self-renewal (Wray *et al*, 2010; Dunn *et al*, 2014) (Appendix Fig S9A). Both null and MLS-Stat3 cultures underwent differentiation and cell death and collapsed completely within three passages. In contrast, a *Stat3* null clone transfected with wild-type Stat3 (clone B, see Fig 5B) displayed robust self-renewal in LIF + PD and expression of nuclear Stat3 targets (Appendix Fig S9A and B).

Thus, Stat3 specifically localized to the mitochondria is able to enhance transcription of mitochondrial genes and proliferation, but is unable to sustain ES cell identity.

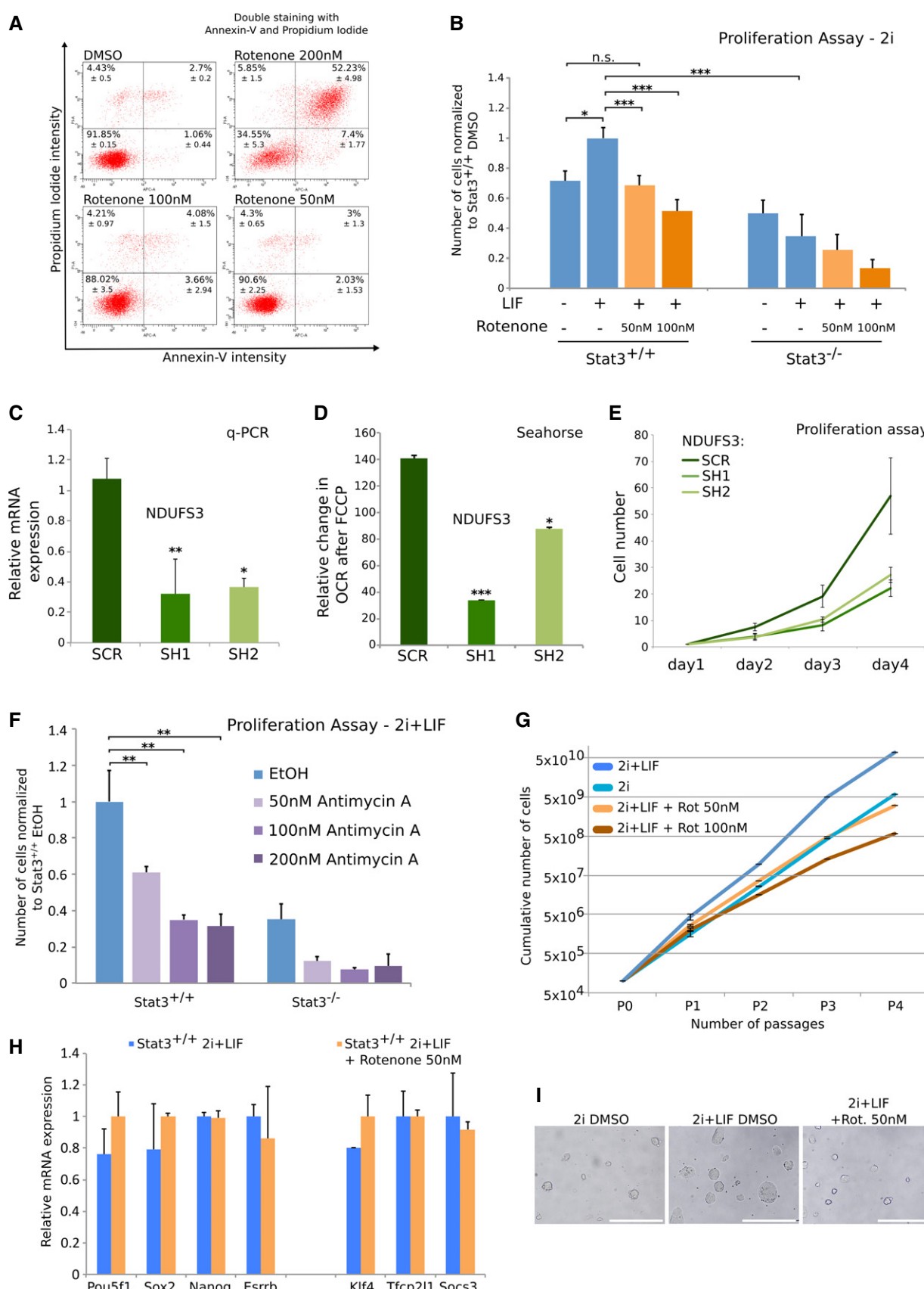

Figure 4.

**Figure 4.   LIF promotes proliferation via respiration.**

A　Flow cytometry analysis after double staining with annexin V and propidium iodide in ES cells. The combination of annexin V-FITC and propidium iodide allows the distinction between viable cells (unstained) bottom left quadrants, early apoptotic cells (annexin V-FITC positive) bottom right quadrants, late apoptotic and/or necrotic cells (annexin V-FITC and propidium iodide positive) top right quadrants. ES cells were treated with increasing concentrations of rotenone from 50 nM to 200 nM for 48 h. A slight increase in cell death could be observed only at a concentration of 200 nM of rotenone. In each quadrant, the mean and s.e.m. of three independent experiments are indicated.

B　Proliferation assay of Stat3$^{+/+}$ and Stat3$^{-/-}$ cells cultured in 2i. Cells were seeded and treated for 48 h with LIF and rotenone (orange bars), a Complex I inhibitor, as indicated. Scores were normalized to Stat3$^{+/+}$ cells treated with LIF and DMSO. Proliferation was enhanced by LIF treatment in Stat3$^{+/+}$ cells and reduced by rotenone treatment. Stat3$^{-/-}$ cells are more sensitive to rotenone treatment. Mean and s.e.m. of at least 2 independent experiments are shown. Unpaired t-test: *$P < 0.05$, ***$P < 0.001$. n.s.: non-significant. See also Appendix Fig S4A.

C　Gene expression analysis of Stat3$^{+/+}$ cells transfected with control shRNA (SCR, dark green), and two independent shRNAs for a Complex I subunit (Ndufs3) (SH1 and SH2). Note that shRNAs for Ndufs3 downregulate gene expression of about 70%. Mean and s.d. of two independent experiments are shown. Unpaired t-test: *$P < 0.05$, **$P < 0.01$.

D　Relative changes in oxygen consumption after 200 nM FCCP treatment of Stat3$^{+/+}$ cells transfected with control shRNA (dark green) and two different shRNAs for Ndufs3. Note that downregulation of the Complex I subunit results in decreased respiration. Mean and s.e.m. of > 4 technical replicates are shown. Unpaired t-test: *$P < 0.05$, ***$P < 0.001$. See also Appendix Fig S4D for full Seahorse profiles.

E　Proliferation assay of control shRNA cells (dark green) and cells with downregulation of Ndufs3. Note that downregulation of the Complex I subunit results in decreased proliferation. Mean and s.e.m. of three independent experiments are shown.

F　Proliferation assay of Stat3$^{+/+}$ and Stat3$^{-/-}$ cells cultured in the presence of LIF showing the reduction in proliferation after 48-h treatment with increasing concentrations of antimycin (50, 100, or 200 nM). See also Appendix Fig S4F. Scores were normalized to WT cells treated with a vehicle (EtOH). Mean and s.e.m. of two independent experiments are shown. Unpaired t-test: **$P < 0.01$. See also Appendix Fig S4E.

G　Proliferation assay showing the effect of long-term treatment of rotenone on cell proliferation. Stat3$^{+/+}$ cells cultured in 2i and 2i + LIF with or without rotenone were scored for 4 subsequent passages. The cumulative number of cells has been calculated and shown on log scale. Rotenone at 50 or 100 nM reduced the number of cells after 4 passages by 4.8 and 16.7 times, respectively. Mean and s.d. of two technical replicates of a representative experiment are shown. See also Appendix Fig S6E.

H　Gene expression analysis of Stat3$^{+/+}$ cells cultured in 2i + LIF (blue bars) and with 50 nM rotenone for 48 h (orange bars) showing that rotenone does not affect the transcription of the main LIF targets in ES cells. Data were normalized to the highest value for each condition. Mean and s.d. of 2 biological replicates are shown.

I　Bright field images showing that long-term treatment with rotenone causes a reduction in proliferation, shown by the reduced size of individual colonies, but does not cause differentiation in mES cells. Scale bar, 50 μm.

## LIF-dependent regulation of mitochondrial activity is critical for induction of naïve pluripotency

Signaling from LIF via Stat3 is important for the induction of naïve pluripotency. Over-activation of LIF/Stat3 is sufficient to reprogram EpiSCs to naïve pluripotent iPS cells (Han *et al*, 2010; Yang *et al*, 2010; Onishi *et al*, 2014). Expression of several transcription factors can also convert EpiSCs into naïve iPS cells (Guo *et al*, 2009; Hall *et al*, 2009; Hanna *et al*, 2009; Silva *et al*, 2009; Han *et al*, 2010; Festuccia *et al*, 2012; Gillich *et al*, 2012; Martello *et al*, 2013), but the presence of LIF invariably enhances the efficiency of conversion. The preceding results provoked the suggestion that LIF may exert functions during reprogramming beyond rewiring of the transcription factor network controlling pluripotency. During reprogramming of EpiSCs, mitochondrial respiration must be actively boosted to the level of naïve pluripotent cells (Zhou *et al*, 2012). We hypothesized that LIF could contribute directly by promoting mitochondrial transcription and activity.

We confirmed that EpiSCs have a greatly reduced OCR compared with ES cells (Appendix Fig S10A). We also found a general reduction in expression of known nuclear and mitochondrial Stat3 targets in EpiSCs (Fig 6A). To examine the involvement of mitochondrial respiration in the reprogramming process, we took advantage of the GOF-18 EpiSC line (Han *et al*, 2010). A fraction of GOF18 EpiSCs exhibit spontaneous conversion in 2i after 48 h exposure to LIF (Han *et al*, 2010; Yang *et al*, 2010). When the Complex I inhibitor rotenone was added together with LIF, we observed a severe reduction in the yield of iPS cell colonies (Fig 6B and C). Similar results were obtained in a second EpiSC line (Appendix Fig S10B) in which resetting to naïve pluripotency is driven by transient hyperactivation of Stat3 (Yang *et al*, 2010). We also exposed cells to rotenone 4 days after LIF induction and observed no difference in the number of iPS

colonies obtained (Fig 6D and Appendix Fig S10C). Importantly, the low doses of rotenone used are tolerated well by both EpiSCs and ES cells (Appendix Fig S10D–G and Fig 4A and G), suggesting that the reduction in colony number is not due to toxicity. iPS cells obtained either in the presence or in the absence of rotenone treatment could self-renew in 2i + LIF without feeders over multiple passages and showed reactivation of naïve markers and shutdown of EpiSC markers (Fig 6E), suggesting that they are *bona fide* naïve pluripotent cells. These results suggest that upregulation of mitochondrial respiration is specifically required during the first 4 days of reprogramming.

To elucidate further the molecular mechanism underlying the effect of LIF and rotenone on reprogramming, we inspected the mitochondrial and nuclear transcriptional targets of Stat3 and observed that 48 h of LIF treatment in 2i is sufficient to induce both classes of gene in GOF18 EpiSCs (Fig 6F and G, compare blue and gray bars). The upregulation of mitochondrial targets indicates that Stat3 is active in the mitochondria in EpiSCs. All Stat3 targets are induced at the same levels, either in the presence or in the absence of rotenone (compare gray and purple bars), in agreement with previous results (Fig 4G) that treatment with rotenone does not affect the intensity of LIF signaling. Therefore, we conclude that rotenone acts downstream of LIF signaling without affecting signal transduction.

Our findings suggest that LIF activates two programs, a nuclear program promoting rewiring of the transcription factor network and a mitochondrial program important for resetting the metabolic profile of the cell. To deconvolute the relative contributions of each program, we expressed two critical nuclear targets, Klf4 and Tfcp2l1 (Niwa *et al*, 2009; Martello *et al*, 2013; Ye *et al*, 2013), in EpiSCs. Either factor, or the combination of the two together, was sufficient to reprogram EpiSCs without LIF consistent with previous reports (Yang *et al*, 2010; Martello *et al*, 2013), but in all cases the

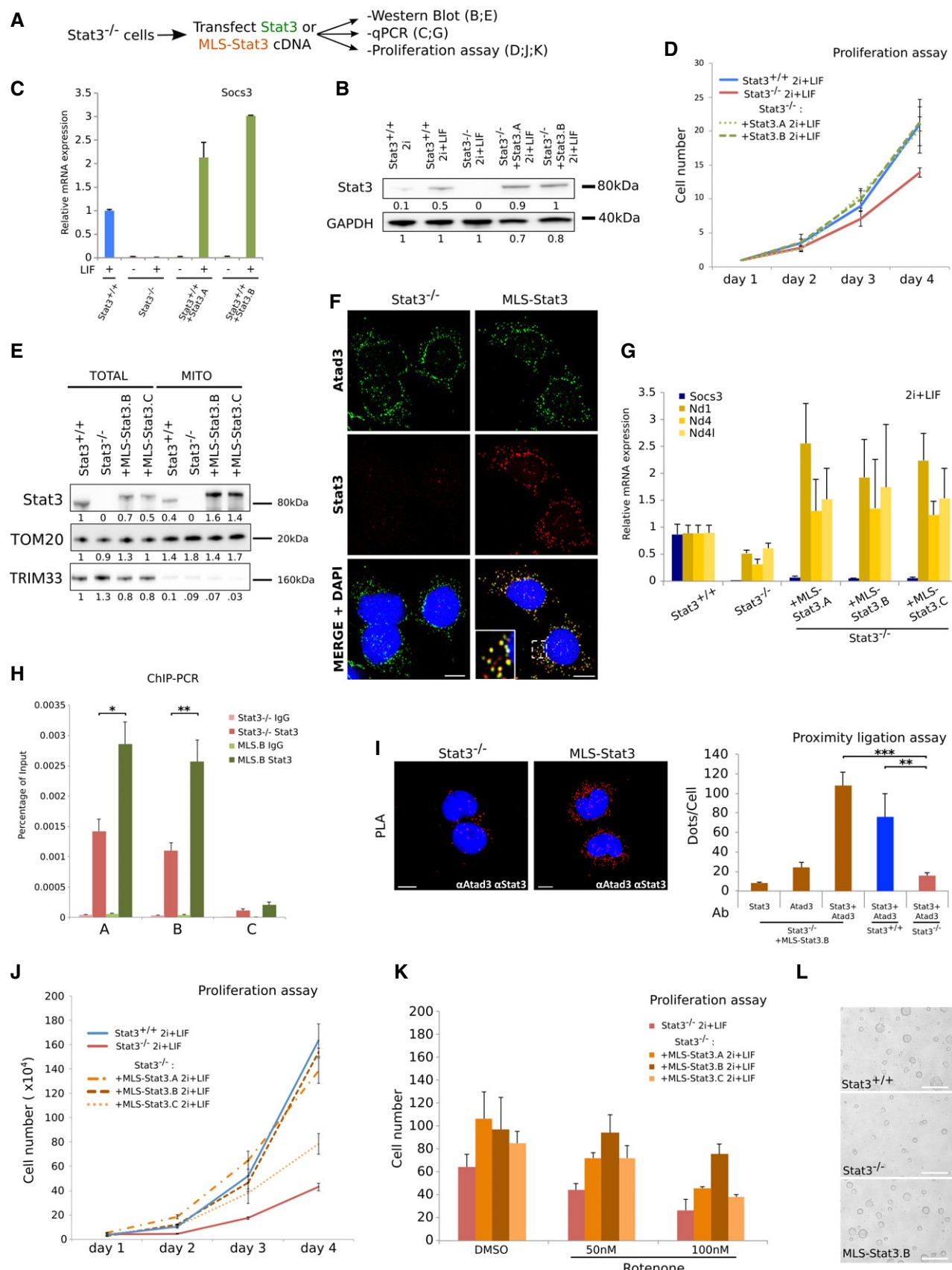

Figure 5.

**Figure 5.  Mitochondrial Stat3 enhances the transcription of mitochondrial genes and proliferation of ES cells.**

A   Experimental approach used to characterize the functional role of Stat3 on cell proliferation and mitochondrial activity.

B   Western blot of Stat3$^{+/+}$ cells cultured in the presence or absence of LIF, Stat3$^{-/-}$ cells cultured in 2i + LIF, and two clones of Stat3$^{-/-}$ cells transfected with a full-length form of Stat3 cultured in 2i + LIF. Relative mean intensity is shown below each band. Note that Stat3 protein levels in clone A and B are comparable to the endogenous levels of the control. GAPDH was used as a loading control.

C   Gene expression analysis of Stat3$^{+/+}$ cells, Stat3$^{-/-}$ cells, and two Stat3 rescue clones (Stat3.A/B) cultured in the absence or presence of LIF. Note that both clones respond to LIF and activate Stat3 direct target Socs3.

D   Proliferation assay of Stat3$^{+/+}$ cells, Stat3$^{-/-}$ cells, and Stat3.A/B rescue clones cultured in the presence of LIF. Cells were seeded and scored for 4 days. Scores were normalized to day 1. Mean and s.e.m. of two independent biological replicates of a representative experiment are shown. See also Appendix Fig S7A.

E   Western blot of total and mitochondrial fractions of Stat3$^{+/+}$, Stat3$^{-/-}$ cells, and two MLS-Stat3 clones cultured in 2i + LIF. The nuclear protein TRIM33 and mitochondrial marker TOM20 confirmed successful mitochondrial isolation. Note that MLS-Stat3 is enriched in the mitochondrial fraction, suggesting correct localization of the fusion protein. See also Appendix Fig S7B.

F   Representative confocal images of Stat3$^{-/-}$ and MLS-Stat3 cells stained with anti-Stat3 and anti-Atad3 antibodies. Merge image shows colocalization between Stat3 and the nucleoids marker Atad3 ($R = 0.72$ for MLS-Stat3 cells, $R = 0.22$ in Stat3$^{-/-}$). DAPI serves as a nuclear counterstain. Scale bar, 10 μm.

G   Gene expression analysis of Stat3$^{+/+}$, Stat3$^{-/-}$ cells, and three MLS-Stat3 clones cultured in the presence of LIF. MLS-Stat3 specifically induces expression of mitochondrial markers with negligible effects on the nuclear target Socs3. Mean and s.d. of two technical replicates. See also Appendix Fig S7E.

H   Chromatin immunoprecipitation (ChIP) performed using anti-Stat3 or a rabbit control IgG antibody in Stat3$^{-/-}$ and MLS-Stat3 cells cultured in 2i + LIF conditions. ChIP-PCR was performed with primers located on three regions of the D-loop (A, B, C). Note that 2 D-loop regions are significantly enriched in MLS-Stat3 compared to Stat3$^{-/-}$ cells. Mean and s.e.m. of three independent experiments are shown. Unpaired *t*-test: *$P < 0.05$, **$P < 0.01$.

I   Left: representative confocal images of Stat3$^{-/-}$ and MLS-Stat3 cells subjected to proximity ligation assay (PLA) by using anti-Stat3 and anti-Atad3 antibodies. DAPI serves as a nuclear counterstain. Red dots indicate spacial proximity between Stat3 and the nucleoids marker Atad3. Right: histogram showing quantification of PLA performed on Stat3$^{+/+}$, Stat3$^{-/-}$, and MLS-Stat3 cells. A number of red dots/cell are plotted. Note that double staining in MLS-Stat3 cells results in increased number or red dots compared to Stat3$^{-/-}$ cells. Mean and s.e.m. of > 15 cells for each sample are shown. Unpaired *t*-test: **$P < 0.01$, ***$P < 0.001$. Scale bar, 10 μm. See also Appendix Fig S7D.

J   Proliferation assay of Stat3$^{+/+}$, Stat3$^{-/-}$ cells, and three MLS-Stat3 clones cultured in the presence of LIF. Cells were seeded and scored for 4 days. Mean and s.e.m. of two technical replicates of a representative experiment are shown. See also Appendix Fig S8A.

K   Proliferation assay of Stat3$^{-/-}$ cells and three MLS-Stat3 clones cultured in the presence of LIF. Cells were seeded and treated with DMSO or with 50 or 100 nM of rotenone for 48 h. MLS-Stat3 clones also proliferated more than Stat3$^{-/-}$ both basally and in the presence of rotenone. Mean and s.e.m. of two technical replicates of a representative experiment are shown.

L   Representative bright field images of Stat3$^{+/+}$, Stat3$^{-/-}$, and one MLS-Stat3 clone cultured in 2i + LIF showing similar morphology, but note smaller colony size for Stat3$^{-/-}$ cells. Scale bar, 100 μm.

Source data are available online for this figure.

colony yield was enhanced by LIF (Fig 6H and I, and Appendix Fig S10H). Addition of rotenone abrogated the positive effect of LIF on reprogramming, independently of cell line or the reprogramming factor used.

These data suggest that during the first 48 h of the reprogramming process, a significant contribution of LIF to the resetting of pluripotent states is via activation of mitochondrial respiration.

## Discussion

In this study, we show that in ES cells LIF via Stat3 induces expression of mitochondrial genes encoding components of the respiratory chain (Figs 1C–E and 2). In so doing, LIF enhances oxidative phosphorylation (Fig 3A–C) and enables rapid proliferation of ES cells (Fig 4B and G). Moreover, our data indicate that LIF stimulation of mitochondrial respiration facilitates efficient resetting of EpiSCs to naïve pluripotency (Fig 6). Strikingly, the effects of the LIF/Stat3 axis on mitochondria are separable from the previously well-characterized role in naïve pluripotent identity (Figs 4H and 6J, and Appendix Fig S9).

Earlier studies showed that Stat3 may localize to the mitochondria and modulate the respiratory activity of somatic cells (Wegrzyn *et al*, 2009; Meier & Larner, 2014). Several mechanisms have been described, such as interaction of Stat3 with Complex I to enhance its activity, or regulation of the permeability transition pore. Furthermore, direct binding of Stat3 protein on mtDNA has recently been reported in keratinocytes (Macias *et al*, 2014). Our results complement and extend these reports, by showing that in ES cells Stat3

increases expression levels of mitochondrial genes. This in turn leads to increased assembly of respiratory chain complexes, ultimately resulting in enhanced respiratory activity.

Our results do not exclude involvement of other mechanisms, such as direct interaction between Stat3 and Complex I, or regulation of mitochondrial transcript stability by Stat3, which could synergize with the transcriptional effects we observe. Indeed, the relative contribution of different mechanisms may vary according to cell type and signaling environment. For instance, Stat1 acts as a repressor of mitochondrial genome transcription and biogenesis (Meier & Larner, 2014). In cancer cells, Stat3 has been shown to act as either a positive or a negative regulator of mitochondrial activity (Demaria *et al*, 2014) depending on specific post-translational modifications. Thus, various parameters may modulate the net effect of JAK/Stat pathway activation on mitochondrial respiration.

The LIF/Stat3 axis appears to control mES cell proliferation by activating mitochondrial respiration. Other nuclear Stat3 targets have been implicated in the regulation of mES cell proliferation, such as Pim1 (Aksoy *et al*, 2007; Bourillot *et al*, 2009). When mitochondrial respiration was blocked by specific inhibitors, we still observed that Stat3$^{-/-}$ cells proliferate less than Stat3$^{+/+}$ (Fig 4B and F, and Appendix Fig S5B). Such decrease in proliferation could be due to reduced expression of Pim1 or other targets in Stat3$^{-/-}$ cells.

Naïve pluripotent cells have a bivalent metabolism, characterized by high levels of both mitochondrial respiration and glycolysis (Zhou *et al*, 2012; Teslaa & Teitell, 2015). Our results show that the LIF/Stat3 axis potentiates mitochondrial respiration. We should stress, however, that in the absence of LIF or Stat3, respiratory

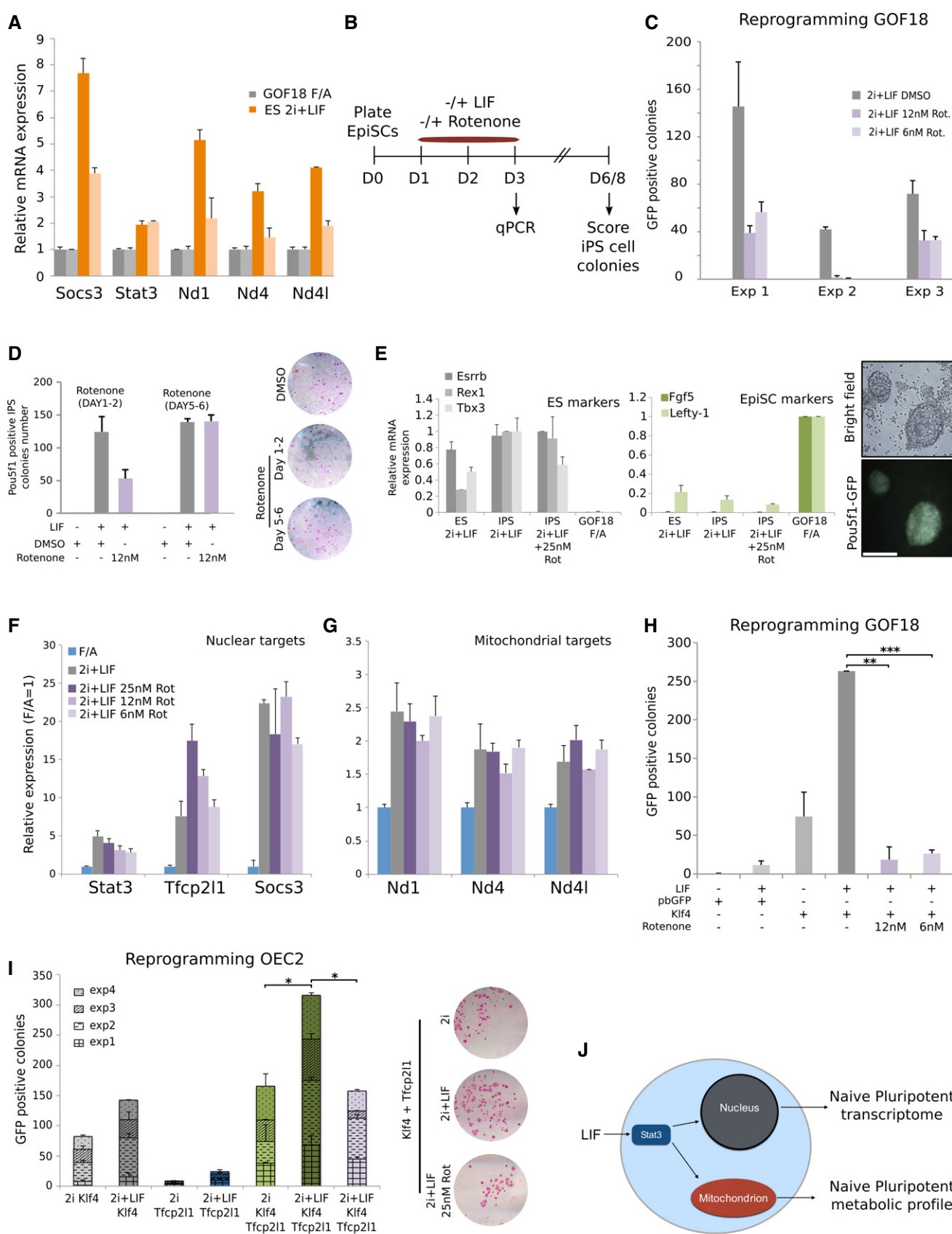

**Figure 6.**

◄

**Figure 6. LIF-dependent mitochondrial activation increases reprogramming efficiency in EpiSC.**

A Gene expression analysis of EpiSCs (GOF18) cultured in FGF + activin (gray bars) and ES cells cultured in 2i + LIF (orange bars) shows a general reduction in both nuclear LIF targets (Socs3, Stat3) and mitochondrial markers (Nd1, Nd4, Nd4 l) in EpiSCs. Data were normalized to EpiSCs in F/A, and mean and s.e.m. of two technical replicates of two independent experiments (light and dark bars) are shown.

B Experimental scheme for testing the effect of rotenone on EpiSCs reprogramming. EpiSCs were plated in FGF + activin (day 0) and cultured for 24 h. Cells were then treated with or without LIF and rotenone at different concentrations as indicated (day 1). Cells were either harvested after 48 h (day 3) for gene expression analysis or left in 2i media, and Oct4-GFP-positive colonies were scored at day 6 or 8.

C Quantification of Oct4-GFP-positive iPS colonies at day 7 of reprogramming generated from GOF18 EpiSCs subjected to 48-h treatment (at D1–D3) with LIF and two doses of rotenone. Note that rotenone treatment reduces the number of iPS colonies generated. Mean and s.e.m. of two technical replicates of three independent experiments are shown. See also Appendix Fig S10B.

D Left: number of Oct4-GFP-positive colonies at day 8 of reprogramming upon 48-h treatment with rotenone at days 1–2 or at days 5–6. Note that rotenone treatment given at days 5–6 does not reduce the number of iPS colonies. Mean and s.e.m. of two technical replicates are shown. See also Appendix Fig S10C. Right: representative alkaline phosphatase (AP) staining of iPSc colonies generated from GOF18 EpiSCs at day 7 of reprogramming treated with DMSO (top) or with rotenone for 48 h at days 1–2 (middle) or 5–6 (bottom).

E Gene expression analysis of iPS cells generated from GOF18 EpiSCs and cultured in 2i + LIF for at least 2 passages. Note that iPS cells show high expression of naïve pluripotency markers (gray bars on left panel) and low expression of EpiSC markers (green bars on right panel). Data were normalized to the highest value for each gene, and mean and s.e.m. of three technical replicates are shown. Right panel shows representative bright field and Pou5f1-GFP$^+$ images of iPS colonies at day 8 of reprogramming. Scale bar, 50 μm.

F, G Gene expression analysis of GOF18 EpiSCs cultured in FGF + activin or treated for 48 h with 2i + LIF in the absence or presence of decreasing concentrations of rotenone. Note that both nuclear (F) and mitochondrial (G) genes expression is induced upon LIF treatment and rotenone did not impair LIF effect. Data were normalized to F/A, and mean and s.e.m. of three independent experiments are shown.

H The number of iPS colonies generated from GOF18 EpiSCs transfected transiently with a piggyBac vector containing GFP or Klf4 and treated with LIF for 48 h and rotenone as indicated. Mean and s.e.m. of two technical replicates of a representative experiment are shown. Unpaired *t*-test: **$P < 0.01$, ***$P < 0.001$. See also Appendix Fig S10H.

I Left panel shows the number of GFP-positive iPS colonies generated from OEC2 EpiSCs transiently transfected with a piggyBac vector containing Klf4 (1$^{st}$ and 2$^{nd}$ bar), Tfcp2l1 (3$^{rd}$ and 4$^{th}$ bar), or a combination of both (5$^{th}$ and 6$^{th}$ bar) and treated with LIF for 48 h or rotenone as indicated. Note that rotenone abolished the effect of LIF on iPS formation. Each bar shows the cumulative number of colonies from four independent experiments, and mean and s.e.m. of two technical replicates for each experiment are shown. Unpaired *t*-test: *$P < 0.05$. Right panel shows representative alkaline phosphatase (AP) staining of iPSc colonies generated from OEC2 EpiSCs at day 7 of reprogramming transfected with a piggyBac vector containing Klf4 and Tfcp2l1 and treated with 2i (top), 2i + LIF (middle) or with 2i + LIF and rotenone for 48 h at days 1–2 (bottom).

J Model depicting the dual role of Stat3 as an inducer of nuclear transcription factors critical for maintenance and induction of naïve pluripotency and, at the same time, as an activator of mitochondrial transcription and activity responsible for the high levels of respiration observed in naïve pluripotent cells.

activity is reduced, but not ablated in ES cells. Interestingly, glycolysis does not appear to be affected by LIF/Stat3 activation. In particular, we did not observe a compensatory increase in glycolysis when mitochondrial respiration was reduced, even though cells are cultured in high concentrations of glucose (Appendix Fig S3B and C). This may suggest that glycolysis serves as the basal energy source and mitochondrial respiration is used as "spare capacity" that can be enhanced by LIF/Stat3.

Activating mitochondrial respiration represents a potential roadblock in the induction of naïve pluripotency. In this context, the transcription factor network controlling pluripotency operates in tandem with metabolic regulation. Thus, when pluripotency factors are expressed in EpiSCs, LIF further enhances reprogramming in a manner dependent on mitochondrial respiration. Conversely, chemical blockade of mitochondrial respiration does not affect the expression of pluripotency markers. This result is consistent with several reports showing that inhibition of LIF signaling, via Jak inhibitor or genetic inactivation of Stat3, potently reduces or abolishes reprogramming to naïve pluripotency, even if critical factors such as Oct4, Sox2, and Klf4 are over-expressed (Yang *et al*, 2010; van Oosten *et al*, 2012; Tang *et al*, 2012).

Our study links the LIF/Stat3 axis directly to mitochondrial activity, but we still do not know how the levels of mitochondrial respiration affect ES cell proliferation or EpiSC resetting. Preliminary evidence would suggest that this is not simply due to altered ATP production. It will be interesting to investigate how LIF/Stat3 affects the global metabolic profile of ES cells in order to identify specific pathways involved in increased proliferation. Several recent studies have highlighted the potential crosstalk between

metabolism and epigenetic modifications in pluripotent cells (Lu & Thompson, 2012; Carey *et al*, 2015). The action of LIF/Stat3 on mitochondria may therefore be important not to fulfill a bioenergetic requirement, but to increase the production of cofactors for epigenetic processes, such as demethylation of DNA and histone modification.

# Materials and Methods

### Embryonic stem cell culture

ESCs were cultured without feeders on plastic coated with 0.2% gelatine (Sigma, cat. G1890) and replated every 3–4 days at a split ratio of 1:10 following dissociation with Accutase (GE Healthcare, cat. L11-007). Cells were cultured either in serum-free N2B27-based medium (DMEM/F12 and Neurobasal [both Life Technologies] in 1:1 ratio, 0.1 mM 2-mercaptoethanol, 2 mM L-glutamine, 1:200 N2 [Life Technologies], and 1:100 B27 [Life Technologies]) supplemented with small-molecule inhibitors PD (1 μM, PD0325901), CH (3 mM, CHIR99021) from Axon (cat. 1386 and 1408) and LIF (100 units/ml produced in house), or in GMEM (Sigma, cat. G5154) supplemented with 10% FBS (Sigma, cat. F7524), 100 mM 2-mercaptoethanol (Sigma, cat. M7522), 1× MEM non-essential amino acids (Invitrogen, cat. 1140-036), 2 mM L-glutamine, 1 mM sodium pyruvate (both from Invitrogen), and 100 units/ml LIF.

EpiSCs were cultured without feeders on plastic coated with fibronectin (Millipore, cat. FC010) and replated every 2 days at a

split ratio of 1:10 following dissociation with Dispase (Stem Cell Technologies, cat. 07923). Cells were cultured in serum-free media N2B27 (see above) supplemented with FGF2 (12 ng/ml) and activin (20 ng/ml) produced in house. Oct4-GFP (OEC2 Y118 line) was described in Yang *et al* (2010). GOF18 EpiSCs were described in Han *et al* (2010) and generously provided by Hans Schöler.

For DNA transfection, we used Lipofectamine 2000 (Life Technologies, cat. 11668-019) and performed reverse transfection. For one well of a 6-well plate, we used 6 μl of transfection reagent, 2 μg of plasmid DNA, and 300,000 cells in 2 ml of N2B27 medium. The medium was changed after overnight incubation.

Stable transgenic ESCs lines expressing Stat3 or MLS-Stat3 were generated by transfecting cells with PiggyBac transposon plasmids CAG-Stat3 or CAG-MLS-Stat3 with piggyBac transposase expression vector pBase. Selection for transgenes was applied, and stable clones were selected in 2i conditions.

For LIF induction experiments, ES cells were cultured in 2i without LIF for > 2 passages, plated (8,000 cells/cm$^2$) in 2i. Twenty-four hours after plating, cells were treated with LIF for the indicated amount of time.

For AP staining, cells were fixed with a citrate–acetone–formaldehyde solution and stained using the Alkaline Phosphatase kit (Sigma, cat. 86R-1KT). Plates were scanned using a Nikon Scanner and scored manually.

### RNA-seq data analysis

RNA sequencing data used in this study are described in Martello *et al* (2013) and are available in the ArrayExpress repository under accession E-MTAB-1796.

### Proliferation assay

Cell proliferation was assessed by plating 15,000 ES cells in 12-well plate. Cells were counted every 24 h for 4 days. For rotenone (Sigma, cat. R8875), antimycin A (Sigma, cat. A8674), and myxothiazol (Sigma, cat. T5580) treatments, cells were plated in the presence of the inhibitors and scored after 48 h.

### Cell cycle analysis

Cell cycle analysis was performed by staining single live cells with propidium iodide (Sigma, cat. P4170), according to the manufacturer's instructions. Samples were analyzed by flow cytometry using a BD FACSCanto$^{TM}$ cytometer.

### ROS measurement assay

Reactive oxygen species production was detected by staining single live cells with 2′,7′-dichlorodihydrofluorescein diacetate (H2DCFDA) (Life Technologies, cat. D399), according to the manufacturer's instructions. Samples were analyzed by flow cytometry using a BD FACSCanto$^{TM}$ cytometer.

### Reprogramming assay

EpiSCs lines bearing an Oct4-GFP reporter were plated in Fgf2/activin medium and switched to 2i + LIF (with or without rotenone)

conditions the next day. Human LIF was used at a concentration of 20 ng/ml. Cells were kept in 2i + LIF medium for 5 days, if not indicated otherwise, before switching to 2i. Reprogramming experiments were ended 6/8 days after medium switch to 2i/LIF, and Oct4-reporter-positive iPSc colonies were scored manually.

### Propidium iodide/annexin V staining

PI/AnnV staining was performed on live single ESCs or EpiSCs according to the manufacturer's instructions (Ebioscience, ref. 88-8007-72). Samples were analyzed by flow cytometry using a cytometer BD FACSCanto$^{TM}$ with BD FACSDiva$^{TM}$ software.

### Flow cytometry

After treatment with Accutase (GE Healthcare, cat. L11-007), dissociated ESCs were resuspended in PBS. Flow cytometry analyses were performed using a cytometer BD FACSCanto$^{TM}$ with BD FACSDiva$^{TM}$ software.

### Mitochondria isolation

Mitochondria isolation was performed from $4 \times 10^7$ cells pellet as previously described in Frezza *et al* (2007). For BNGE analysis, mitochondria were isolated as described in Cogliati *et al* (2013).

### Oxygen consumption assay

Oxygen consumption was measured using the Seahorse XF24 (Seahorse Bioscience). For this, ~20 h before the analysis Stat3$^{+/+}$ and Stat3$^{-/-}$ cells were seeded in a 24-well cell culture plate (Seahorse Bioscience) coated with laminin (Sigma, cat. L2020) at a density of 140,000 cells per well in N2B27 media supplemented with 2i or 2i + LIF (as indicated). It is crucial to have an evenly plated monolayer of cells to obtain reliable measurements. Cells were maintained in a 5% $CO_2$ incubator at 37°C, and 1 h before the experiment, the cells were washed and incubated in 600 μl of DMEM containing 10 mM glucose (DMEM-high glucose) pH 7.4 at 37°C in a non-$CO_2$ incubator.

During the experiment, oxygen concentration was measured over time periods of 2 min at 5-min intervals, consisting of a 3-min mixing period and 2-min waiting period. Measurements of OCR in basal conditions were used to calculate the basal mitochondrial respiration. After this, the mitochondrial uncoupler FCCP (carbonyl cyanide-p-trifluoromethoxyphenylhydrazone) was added into the media at a final concentration of 200 nM. Oxygen consumption during this phase reflects the maximal mitochondrial respiratory capacity. Finally, ETC activity was blocked by the addition of rotenone or antimycin A, both at a final concentration of 200 nM. As a result, OCR drops dramatically, and the oxygen consumed in this situation by the cells comes from a non-mitochondrial origin.

### Gene expression analysis by quantitative PCR with reverse transcription

Total RNA was isolated using RNeasy kit (QIAGEN), and complementary DNA (cDNA) was made from 1 μg using M-MLV Reverse

**Table 1. PCR primers.**

| Gene | Forward primer sequence | Reverse primer sequence |
|---|---|---|
| mNd1 | ccattctaatcgccatagcc | atgccgtatggaccaacaat |
| mNd4 | cgcctactcctcagttagcc | gtgaggccatgtgcgattat |
| mNd4l | ctccaactccataagctcca | ggctgcgaaaactaagatgg |
| mCo3 | taacccttggcctactcacc | ataggagtgtggtggccttg |
| mPouf5 | gttggagaaggtggaaccaa | ctccttctgcagggctttc |
| mSox2 | cacaactcggagatcagcaa | tctcggtctcggacaaaagt |
| mNanog | ttcttgcttacaagggtctgc | agaggaagggcgaggaga |
| mEsrrb | ggcgttcttcaagagaacca | cccactttgaggcatttcat |
| mKlf4 | cgggaagggagaagacact | gagttcctcacgccaacg |
| mTfcp2l1 | ggggactactcggagcatct | ttccgatcagctcccttg |
| mSocs3 | atttcgcttcgggactagc | aacttgctgtgggtgaccat |
| mStat3 | tgttggagcagcatcttcag | gaggttctccaccaccttca |
| mRex1 | tcttctctcaatagagtgagtgtgc | gctttcttctgtgtgcagga |
| mFgf5 | aactccatgcaagtgccaaat | cggacgcataggtattatagctg |
| mLefty1 | ccaaccgcactgcccttat | cgcgaaacgaaccaacttgt |
| mbactin | ctaaggccaaccgtgaaaag | accagagggcatacagggaca |
| mNdufS3 | ttatggcttcgagggacatc | attcttgtgccagctccact |
| ChIP A | cattaaactattttccccaagca | caaatggggaaggggatagt |
| ChIP B | aaatgcgttatcgcccatac | tcttcaccgtaggtgcgtct |
| ChIP C | tagtccgcaaaacccaatca | ttgatcaggacatagggtttga |

**Table 2. Antibodies.**

| Antibody | Species | Source | Dilution |
|---|---|---|---|
| anti-Stat3 | Mouse monoclonal | Cell Signalling cat. 9139 | WB: 1:1,000 IF: 1:100 |
| anti-PStat3 (Y705) | Rabbit monoclonal | Cell Signalling cat. 91455 | WB: 1:2,000 |
| anti-TOM20 | Rabbit polyclonal | Santa Cruz Biotechnologies cat. 11415 | WB: 1:2,000 IF: 1:100 |
| anti-TIMM23 | Mouse monoclonal | BD Biosciences cat. 611223 | WB: 1:1,000 |
| anti-TRIM33 | Mouse monoclonal | Santa Cruz Biotechnologies cat. 101179 | WB: 1:1,000 |
| anti-laminB | Goat polyclonal | Santa Cruz Biotechnologies cat. 6216 | WB: 1:1,000 |
| anti-NDUFB8 | Mouse monoclonal | Abcam cat. AB110242 | WB: 1:1,000 |
| anti-ATP synthase | Mouse monoclonal | Abcam cat. AB14748 | WB: 1:1,000 |
| anti-GAPDH | Mouse monoclonal | Millipore cat. MAB374 | WB: 1:1,000 |
| anti-SDHA | Mouse monoclonal | Abcam cat. Ab14715 | WB: 1:2,000 |
| anti-Atad3A | Rabbit monoclonal | AB-Biotechnologies cat. 224485 | IF 1:100 |
| anti-DNA | Mouse monoclonal | Progen cat. 61014 | IF: 1:1,000 |

Transcriptase (Invitrogen) and dN6 primers. For real-time PCR, we used SYBR Green Master mix (Bioline. Cat. BIO-94020). Primers are detailed in Table 1. Technical replicates were carried out for all quantitative PCR. An endogenous control (beta-actin) was used to normalize expression.

## Luciferase assay

Luciferase reporter plasmid was derived by subcloning of the D-loop promoter region into pGL3-basic luciferase plasmid (Addgene). CMV-lacZ has been previously described in Lukas *et al* (1997).

Embryonic stem cells and EpiSCs were plated in a 12-well plate and transiently transfected with luciferase reporter plasmid with CMV-lacZ to normalize for transfection efficiency (based on CPRG (Merck) colorimetric assay), together with plasmids encoding for the indicated proteins. We transfected 1.5 μg of DNA in each sample by adding the pKS Bluescript plasmid when needed. Forty-eight hours after transfection, the cells were harvested in Luc lysis buffer (25 mM Tris pH 7.8, 2.5 mM EDTA, 10% glycerol, 1% NP-40). Luciferase activity was determined in a Tecan plate luminometer with freshly reconstituted assay reagent (0.5 mM D-luciferin, 20 mM tricine, 1 mM $(MgCO_3)_4 \cdot Mg(OH)_2$, 2.7 mM $MgSO_4$, 0.1 mM EDTA, 33 mM DTT, 0.27 mM CoA, 0.53 mM ATP).

## Immunoblotting

Immunoblotting was performed as previously described in Yang *et al* (2010). For BNGE, immunoblotting was performed as in Cogliati *et al* (2013). For antibodies details, see Table 2. Images

were digitally acquired using a ImageQuant LAS4000 (GE Healthcare).

## Immunofluorescence

For immunofluorescence, cells were fixed for 10 min in cold methanol at −20°C, washed in TBS, permeabilized for 10 min with TBST + 0.3% Triton X-100 at RT, and blocked for 45 min in TBS + 3% goat serum at RT. The cells were incubated overnight at 4°C with primary antibodies. After washing with TBS, the cells were incubated with secondary antibodies (Alexa, Life Technologies) for 30 min at RT.

Cells were mounted with ProLong® Gold Antifade Mountant with DAPI (Life Technologies, cat. P36941). Images were acquired with a Leica SP2 confocal microscope equipped with a CCD camera. For antibodies used, see Table 2. We quantified the degree of colocalization between different proteins by calculating the Pearson's coefficient R by using the "coloc2" function of the freely available software Fiji (http://fiji.sc/Fiji).

## Proximity ligation assay (PLA)

Proximity ligation assay was performed after an overnight incubation with primary antibodies following the manufacturer's instructions (OLink Bioscience). Images were acquired with a Leica SP5 confocal microscope equipped with a CCD camera. Images acquired were analyzed using a custom macro for ImageJ, allowing automated and unbiased analysis.

 

## Chromatin immunoprecipitation (ChIP)

For ChIP experiments, cells were crosslinked, lyzed, and sonicated as described in Enzo *et al* (2015). For immunoprecipitation, sheared chromatin from $5 \times 10^6$ cells was incubated overnight at 4°C with 3 μg of rabbit monoclonal anti-Stat3 (Santa Cruz Biotechnologies, cat. sc-482) or with control rabbit IgG. Protein A Dynabeads (Life Technologies) were added for 3 h after extensive blocking in 0.5% BSA. Washing, de-crosslinking, and DNA purification were performed as in Enzo *et al* (2015). Results were analyzed by qPCR. Since the D-loop region is partially duplicated in the nuclear genome, we designed primers specific for the mitochondrial genome (see Table 2).

## Data availability

Stat3 RNA sequencing data: Martello *et al* (2013). Stat3 ChIP sequencing data: Chen *et al* (2008).

**Expanded View** for this article is available online.

## Acknowledgements

We thank members of the Smith laboratory and the Martello laboratory for advice and discussion. We are grateful to Sirio Dupont, YaoYao Chen and Christian Frezza for critical reading of the manuscript and to Irene Zorzan, Marco Sciacovelli, and Valentina Giorgio for technical support. GM's laboratory is supported by grants from Giovanni Armenise-Harvard Foundation and Telethon Foundation (TCP13013). The Cambridge Stem Cell Institute receives core funding from the Wellcome Trust and Medical Research Council. GM was supported by a Human Frontier Science Program Fellowship. AS is a medical research professor.

## Author contributions

GM and AGS designed the study. GM, EC, and RMB carried out, analyzed, and interpreted experiments. MES performed BNGE assays, provided reagents and technical support. GM and AS supervised the study and wrote the paper.

## Conflict of interest

The authors declare that they have no conflict of interest.

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
