## [Review Process File · The EMBO Journal]

Manuscript EMBO-2015-92629

LIF/Stat3 directly promotes mitochondrial respiration during maintenance and induction of naive pluripotency.

Elena Carbognin, Mr. Riccardo Betto, Maria Soriano, Austin Smith and Graziano Martello

Corresponding author: Graziano Martello, University of Padova

Review timeline:

Submission date:	21 July 2015
Editorial Decision:	26 August 2015
Revision received:	14 December 2015
Editorial Decision:	12 January 2016
Revision received:	14 January 2016
Accepted:	15 January 2016

Editor: David del Alamo

Transaction Report:

1st Editorial Decision

26 August 2015

Thank you again for the submission of your manuscript entitled "LIF/Stat3 directly promotes mitochondrial respiration during maintenance and induction of naive pluripotency" and please accept my apologies for the delay in responding due to the summer break. I have now received the reports from the two referees that accepted to evaluate your manuscript, which I copy below.

As you can see from their comments, both referees are to a certain extent supportive of the publication of your study in The EMBO Journal, but point out to a number of significant concerns that will require considerable additional experimental work before we can accept your manuscript. I will not repeat here the referee concerns, which we believe, are clear, as they are their suggestions for improvement of the manuscript. I would like to point out however, that according to both referees, and particularly to referee #2, a conclusive demonstration of the transcriptional activity of STAT3 inside the mitochondria will be required for acceptance of your study. Please contact me if you have any questions, need further input on the referee comments or if you anticipate any problems.

Given the extensive revision that needs to be undertaken, publication of the study in The EMBO journal cannot be warranted at this stage. However, as both referees state the potential interest of your findings, I would like to invite you to address the reviewers concerns with the understanding that they must be fully addressed.

Thank you very much again for the opportunity to consider your work for publication. I look forward to your revision.

REFEREE REPORTS

Referee #1:

This work advances the study of the role of STAT3 in mitochondria. The system is stem cells; and the relevance is twofold. First, there is direct interest in the mechanism of STAT3 in relation to maintenance of pluripotency. Second, there is potential insight into the mechanism of pluripotency to metabolism. Lastly, there is a potentially novel mechanism of STAT3 mediated regulation of mitochondrial function. The paper is of good technical quality. There are several issues in need of major clarification, however.

- The use of rotenone and antimycin A are likely confounded by the production of reactive oxygen species, either beneficial (signaling) or cytotoxic. The extent of ROS production by the use of these inhibitors should be clarified. To confirm the relationship to complex I inhibition, a knockdown of a nuclear encoded subunit of complex I should also be studied. Furthermore, a complex III inhibitor that does not increase ROS production, at least from complex III, such as myxothiazole or stigmatellin, should be studied.

- There is a poorly defined statement of STAT3-induced induction of mitochondrial DNA transcription independent of TFAM. Evidence of this process should be provided.

- It remains difficult to observe which changes are significant and which are not based upon the current presentation of data.

Referee #2:

LIF/Stat3 signaling has been shown previously to maintain the undifferentiated state of mouse pluripotent stem cells (mPSCs), with Stat3 functioning as a transcription factor in the nucleus. Now, Carbognin and colleagues investigate mechanisms that maintain (or increase during reprogramming) respiration in naïve mPSCs and report that LIF/Stat3 signaling increases steady-state mitochondrial transcripts encoding respiratory chain components which enhances respiration and support rapid self-renewal.

Summary:

The manuscript is well written on an interesting topic. I have some concerns not so much for the effects being reported but for the underlying reason given for these effects. Prior studies have shown no role for Stat3 in mtDNA transcription and, rather, a role as an interacting protein in ETC respiratory complexes.

In terms of novelty, others have shown that (1) LIF/Stat3 regulates/maintains rapid naïve mPSC self-renewal and (2) that Stat3 has a role as a TF in the nucleus and a less clear role in mitochondria, albeit in different cell contexts from pluripotent mPSCs. A role for LIF/Stat3 has also been shown before for increasing reprogramming efficiency to naïve mPSCs, but a timed role for rotenone in disrupting ETC function and its effect on reprogramming efficiency is new. So the work here helps to connect current knowledge and adds new details to fill in and confirm the gaps and advances understanding for how metabolism is regulated in the naïve mPSC state.

For this reviewer, with many papers showing non-transcriptional effects of Stat activity on mitochondrial function, and the extreme difficulty in studying mitochondrial transcripts, their processing, stability, and the factors that affect these processes, the bar is necessarily high to claim a role for Stats in direct regulation of mtDNA transcription without some convincing, direct evidence.

Specific issues include:

1. Figure 1D- Stat3^{-/-2i} +LIF naïve mPSCs show almost uniform repression between 25-50% for the 13 protein coding mtDNA transcripts, which would likely repress respiration compared to Stat3^{-/- 2i}

naïve mPSCs, given that 2-3 fold transcript inductions elevate respiration for Stat3^{+/+2i} +LIF naïve mPSCs. However, the rate of proliferation (Figure 1A) for Stat3^{-/-2i} naïve mPSCs with or without LIF is about the same, instead of being repressed. If respiration is essential for rapid naïve mPSC proliferation, why would this be the case? Does the extent of mtDNA transcript repression not affect the level of respiration at these time points or at later time points (e.g. day 4 or day 5 for Nd4 and Co3 in Figure 1E)?

2. Figure 1F- the Stat3 reporter assay determines the effect for Stat3 transfection on the D-loop reporter region outside of the mitochondria- it is not shown that Stat3 engages the D-loop to (possibly) directly regulate mtDNA transcription within nucleoid structures located within the mitochondrial matrix. Prior work on Stat3 in the mitochondria of differentiated cells showed its interaction with proteins in ETC complex I and activities independent of any role as an mtDNA transcription factor (Wegrzyn, 2009 Zouein, 2014). The data here does not show that Stat3 has a direct role on mtDNA transcription (and is even at odds with prior findings). Also, Figure 4G studies show that some endogenous Stat3 is present in the mitochondrial fraction of naïve mPSCs with 2i+LIF, but there is no 2i only assessment to know whether Stat3 is in mitochondria because of LIF related signaling or whether some Stat3 is always in the mitochondria in naïve mPSCs.

3. Figure 2A- going back to the proliferation results, how would the OCR profile appear for Stat3^{+/+2i} and Stat3^{-/-2i} naïve mPSCs? These should appear similar to Stat3^{-/-2i} + LIF and not Stat3^{+/+2i} + LIF naïve mPSCs if the extent of respiration has a role in the rate of proliferation, a main thesis of the current study.

4. Figure 2D- The number of copies of mtDNA can vary with the mitochondrial protein biomass inside cells, so it is not clear whether a semi-quantitative measurement of mtDNA copy number means much as a surrogate for the number of mitochondria (or the overall network biomass) or for mitochondrial biogenesis.

5. Page 5 - rationale for focusing on complex I abundance- this makes sense if direct Stat3 regulation of the mtDNA transcriptome is the underlying cause (see point 2 above), but one could also argue that the ultimate electron donating complex, IV, is just as compelling. Furthermore, complex I is the least well defined and understood of all the ETC complexes in the respiratory chain and can have variable components dependent upon cell type and state. Also, since Stat3 has already been shown to interact in one experimental setting with complex I proteins (Wegrzyn, 2009), this and not affects on mtDNA transcription could be the source of reduced complex I levels or supercomplex formation (i.e. enhanced instability and/or turnover). How does complex II abundance appear, since it is entirely nucleus encoded?

6. Figure 2F- The blue native gels that form the basis for supercomplex comparisons should be shown in the supplement.

7. Text page 5 and Figure S3A- the claim is that rotenone decreased OCR by >85% but the graph and legend states a 70% reduction- need to make this consistent.

8. Figure 3- why is efficient respiration important for naïve mPSC proliferation? Is it because ATP is generated (FCCP results in Figures 2A, 2B suggest coupling for the ETC) or because ROS is suppressed (complex inhibitors like rotenone increase ROS)? Or because respiration is linked through ETC complex II to the TCA cycle? Also, rotenone and antimycin target different ETC complexes and have different effects and toxicities, so the comment on bottom of page 5 that (paraphrasing) antimycin was used to rule out off target rotenone effects (assume off-ETC effects is what the authors' mean here) does not make much sense even with both having on-target effects only for the ETC.

9. Figure 3D- what are the long term consequences of rotenone on cell viability? In the 48h assays at the doses used it looked OK, but what about at P3 or P4? These drugs do not wash out and they may accumulate over time and by P4 there are ~20 to 100-fold less cells in the rotenone treated wells than the 2i+LIF only wells- is cell death a part of this slow down?

10. Text comment page 6- the authors' claim transfected Stat3 is "almost at endogenous levels" for Figure 4B, in which their levels are ~2-fold more than endogenous, whereas the claim is that Stat3

+LIF "significantly" elevated mtRNA levels in Figure 1D when these levels are for many genes ~2 fold over uninduced levels. It seems to this reviewer that 2-fold protein overexpression could have significant unintended effects on cell physiology when compared to 2-fold changes in steady-state RNA expression levels.

11. Figure 4E- the claim is that the Stat3 rescued clones showed reduced sensitivity to rotenone, but the ratio of reduction at 50nM and 100nM rotenone for the Stat3^{-/-} control and especially Stat3.A look similar, meaning a similar effect at a different baseline for these two lines. The same is seen for Figure 4J in which the slope between 50nM and 100nM rotenone is the same whether the cells are Stat3^{-/-} or the 3 MLS-Stat3 rescue clones, indicating that rotenone is having the same dose-dependent activity just from different starting points.

12. Figure 4F- Most mitochondria targeted proteins are translocated coupled to their translation on or near the mitochondria and they are translocated in a chaperone-bound unfolded configuration to get across the outer (and inner) mitochondrial membranes. If MLS-Stat3 is shipped into the mitochondria, and endogenous Stat3 is usually phosphorylated by JAKs linked to proline-rich regions of intracellular domains of cytoplasmic membrane receptors, then how does MLS-Stat3 become phosphorylated? Is JAK also in/on mitochondria? Does this mean that some MLS-Stat3 is not unfolded and localized to mitochondria?

13. The claim throughout is that Stat3 directly supports transcription of the mtDNA. However, this is not directly shown and all assays used to assess mtDNA transcripts do not consider that Stat3 could enhance mitochondrial transcript stability, which is a very complex and active area of investigation (e.g. how are mtDNA transcripts turned over?). Again, the only published reports on Stat3 in mitochondria, albeit in different cellular contexts, do not show a role in mtDNA transcription for enhanced mitochondrial function, even in all of the papers the authors' cite to support or imply this role.

14. Figure 4I- what is the MLS-Stat3 + 2i naïve mPSC proliferation rate for the 3 clones shown without LIF?

15. The claim is that LIF increases the efficiency of energy production to support rapid naïve mPSC proliferation but there are no measurements to compare ATP levels in any of the conditions tested that manipulate LIF, Stat3, or the ETC. See point 8 above also....

1st Revision - authors' response

14 December 2015

Response to reviewers

We thank the referees for the interest they expressed in our work and for their constructive and well informed comments.

By addressing the referees' points we believe we have clarified some critical aspects and strengthened our conclusions.

Below is a point-by-point response.

Referee #1:

This work advances the study of the role of STAT3 in mitochondria. The system is stem cells; and the relevance is twofold. First, there is direct interest in the mechanism of STAT3 in relation to maintenance of pluripotency. Second, there is potential insight into the mechanism of pluripotency to metabolism. Lastly, there is a potentially novel mechanism of STAT3 mediated regulation of mitochondrial function. The paper is of good technical quality. There are several issues in need of major clarification, however.

The use of rotenone and antimycin A are likely confounded by the production of reactive oxygen species, either beneficial (signaling) or cytotoxic. The extent of ROS production by the use of these inhibitors should be clarified. To confirm the relationship to complex I inhibition, a knockdown of a nuclear encoded subunit of complex I should also be studied. Furthermore, a complex III inhibitor

that does not increase ROS production, at least from complex III, such as myxothiazole or stigmatellin, should be studied.

We agree that ROS production might be affected by rotenone and antimycin A and could influence the biology of ES cells.

We therefore measured ROS production by dichlorofluorescein diacetate (DCFDA) staining and flow cytometry and found no significant increase in ROS after treatment with either rotenone or Antimycin A (Figure S5A-C) at the doses used in our biological assays. We therefore find no evidence that the negative effects of these inhibitors on ES cell proliferation could be due to altered ROS production. Furthermore, as suggested by the reviewer we tested the effect of myxothiazol on ES cells. Interestingly we observed that at very low concentration it reduced proliferation significantly (Figure S4F), also without increasing ROS production (Figure S5D).

From this set of experiments we conclude that blocking the mitochondrial respiratory chain at different levels (i.e. Complex I or III) consistently results in reduced proliferation, without affecting ROS production.

Referee #1 also suggested performing a knockdown experiment to reduce the activity of Complex I in ES cells. We designed shRNA targeting the nuclear mRNA NDUFS3, coding for a Complex I subunit previously shown to be required for Complex I activity in mouse fibroblasts (Lapiente-Brun et al., 2013). We generated stable ES cell lines where NDUFS3 mRNA was reduced by >60% (two independent shRNA sequences, Figure 4C). Respiratory capacity was reduced in both lines (Figure 4D and S4C). We also observed significantly reduced proliferation compared to cells expressing a control shRNA (Figure 4E). The magnitude of the effect was comparable to that observed after treatment with inhibitors of the mitochondrial respiratory chain.

There is a poorly defined statement of STAT3-induced induction of mitochondrial DNA transcription independent of TFAM. Evidence of this process should be provided.

We have now reworded this statement to say:

“LIF/Stat3 could enhance mitochondrial transcription indirectly, via induction of known mitochondrial master transcriptional regulators, such as PGC-1 or TFAM. Inspection of the RNA-seq data from LIF stimulation showed no induction of either of these regulators (Figure S1C).” We also specifically assayed TFAM expression in the context of MLS-Stat3 expression and found no consistent change (Figure S6E).

- It remains difficult to observe which changes are significant and which are not based upon the current presentation of data.

We reorganised the presentation of some of the results and now include statistical tests in order to make the significance of the results apparent.

Referee #2:

LIF/Stat3 signaling has been shown previously to maintain the undifferentiated state of mouse pluripotent stem cells (mPSCs), with Stat3 functioning as a transcription factor in the nucleus. Now, Carbognin and colleagues investigate mechanisms that maintain (or increase during reprogramming) respiration in naïve mPSCs and report that LIF/Stat3 signaling increases steady-state mitochondrial transcripts encoding respiratory chain components which enhances respiration and support rapid self-renewal.

Summary:

The manuscript is well written on an interesting topic. I have some concerns not so much for the effects being reported but for the underlying reason given for these effects. Prior studies have shown

no role for Stat3 in mtDNA transcription and, rather, a role as an interacting protein in ETC respiratory complexes.

In terms of novelty, others have shown that (1) LIF/Stat3 regulates/maintains rapid naïve mPSC self-renewal and (2) that Stat3 has a role as a TF in the nucleus and a less clear role in mitochondria, albeit in different cell contexts from pluripotent mPSCs. A role for LIF/Stat3 has also been shown before for increasing reprogramming efficiency to naïve mPSCs, but a timed role for rotenone in disrupting ETC function and its effect on reprogramming efficiency is new. So the work here helps to connect current knowledge and adds new details to fill in and confirm the gaps and advances understanding for how metabolism is regulated in the naïve mPSC state.

For this reviewer, with many papers showing non-transcriptional effects of Stat activity on mitochondrial function, and the extreme difficulty in studying mitochondrial transcripts, their processing, stability, and the factors that affect these processes, the bar is necessarily high to claim a role for Stats in direct regulation of mtDNA transcription without some convincing, direct evidence.

Specific issues include:

1. Figure 1D- Stat3^{-/-}2i +LIF naïve mPSCs show almost uniform repression between 25-50% for the 13 protein coding mtDNA transcripts, which would likely repress respiration compared to Stat3^{-/-} 2i naïve mPSCs, given that 2-3 fold transcript inductions elevate respiration for Stat3^{+/+}2i +LIF naïve mPSCs. However, the rate of proliferation (Figure 1A) for Stat3^{-/-} 2i naïve mPSCs with or without LIF is about the same, instead of being repressed. If respiration is essential for rapid naïve mPSC proliferation, why would this be the case? Does the extent of mtDNA transcript repression not affect the level of respiration at these time points or at later time points (e.g. day 4 or day 5 for Nd4 and Co3 in Figure 1E)?

In Figure 1C we highlight the global effect of acute LIF stimulation through comparison of “2i” vs “2i+LIF 1h” at the transcriptome level.

In previous Figure 1D we showed a bar plot of mitochondrial transcripts encoding subunits of the ETC and data normalised to untreated 2i samples, in order to show the effect of LIF treatment. As a control we showed the effect of LIF in Stat3^{-/-} cells, and found that mitochondrial transcripts were not induced, with some of them actually reduced. This presentation of the RNA-seq data is not optimal for direct comparison across all the samples. Therefore we now present the same data as a heatmap (New figure 1D). From this it is evident that Stat3^{+/+} cells in 2i+LIF show expression of mitochondrial transcripts that is significantly higher than all other samples, in accordance with the proliferation results.

Comparison of Stat3^{-/-} and Stat3^{+/+} cells in 2i does not show any significant change (p-value > 0.05 for all the genes shown in Figure 1D). LIF treatment does lead to a reduction in transcript levels in Stat3^{-/-} cells, but this is less than the increase induced by LIF in wild type cells, as confirmed by the RT-qPCR assay in Figure 1E. Our interpretation is that this reduction is not sufficient to affect already slow cell proliferation, which is likely supported mainly by glycolysis.

2. Figure 1F- the Stat3 reporter assay determines the effect for Stat3 transfection on the D-loop reporter region outside of the mitochondria- it is not shown that Stat3 engages the D-loop to (possibly) directly regulate mtDNA transcription within nucleoid structures located within the mitochondrial matrix. Prior work on Stat3 in the mitochondria of differentiated cells showed its interaction with proteins in ETC complex I and activities independent of any role as an mtDNA transcription factor (Wegrzyn, 2009 Zouein, 2014). The data here does not show that Stat3 has a direct role on mtDNA transcription (and is even at odds with prior findings).

We fully agree that Stat3 was previously shown to interact with Complex I. However, our results are not in conflict with those findings but rather extend them by demonstrating an additional mechanism. We have reworded the statement in the Discussion of the manuscript to make this explicit:

“Several mechanisms have been described, such as interaction of Stat3 with Complex I to enhancing its activity, or regulation of the permeability transition pore. Our results complement and extend these reports, by showing that Stat3 directly induces transcripts from the mitochondrial genome

coding for subunits of the respiratory chain. This in turn leads to increased assembly of respiratory chain complexes, ultimately resulting in enhanced respiratory activity. Direct binding of Stat3 protein on mtDNA was also recently shown in keratinocytes."

In order to test whether Stat3 directly regulates mtDNA transcription within nucleoid structures we carried out the following additional experiments:

- a) Immunofluorescence for ATAD3, a known nucleoid marker validates that ATAD3 is localised on mtDNA in ES cells (Figure S2). We then stained MLS-Stat3 expressing cells for Stat3 and ATAD3 and observed overlap of the two markers, suggesting that Stat3 localises in nucleoids (Figure 5F). As a control we stained for Stat3 and TOM20, a known marker of mitochondrial membrane, and found adjacency but no overlap between the two markers (Figure S6C).*
- b) Proximity Ligation Assay (PLA, Söderberg O. et al, 2006 Nature Methods) is a method for detecting individual pairs of neighbouring proteins in cells. We used PLA to test whether Stat3 and the nucleoid component ATAD3 interact in ES cells. We observed strong signal in both Stat3 +/+ and MLS-Stat3 expressing cells, suggesting that indeed Stat3 is closely associated with this critical nucleoid component (Figure 2F, 5I).*
- c) We analysed available Stat3ChIP-seq data from ES cells. We observed significant Stat3 enrichment in the D-loop region (Figure 2D). We confirmed this result by ChIP-PCR in wild type cells (Figure 2E).*

We also used MLS-Stat3 expressing cells to rule out any contribution from nuclear Stat3 and compared them to Stat3 -/- cells as a stringent control for specificity of the antibody. We again detected significant enrichment of Stat3 on the D-loop region in MLS-Stat3 cells (Figure 5H). This set of experiments, along with the luciferase reporter assay (Figure 2A-C) and transcriptomic analysis (Figure 1C-D) provide strong evidence that Stat3 directly regulates mtDNA transcription within nucleoid structures in mES cells.

Interestingly in a recent study Stat3 has been shown to directly interact with mtDNA in keratinocytes (Macias et al, 2014). We now cite this reference in the discussion.

As stated above, however, our conclusions are fully compatible with Stat3 also interacting directly with Complex I (Wegrzyn, 2009 Zouein, 2014).

Also, Figure 4G studies show that some endogenous Stat3 is present in the mitochondrial fraction of naïve mPSCs with 2i+LIF, but there is no 2i only assessment to know whether Stat3 is in mitochondria because of LIF related signaling or whether some Stat3 is always in the mitochondria in naïve mPSCs.

We agree with this point but face a technical limitation. As shown in Figure 5B, the levels of Stat3 in 2i without LIF are very low. This is also seen at the transcript level measured by RNAseq. In the presence of LIF signalling Stat3 mRNA and protein levels increase, because the Stat3 gene is a direct transcription target of Stat3 itself (Bourillot et al, 2009 – Stem Cells).

We tried to detect Stat3 in mitochondrial fractions from cells in 2i but were unable to detect any signal. We cannot rule out that a minor fraction of Stat3 localises to the mitochondria in 2i, but such fraction is below the detection levels of available assays. In contrast, mitochondrial Stat3 can readily be detected in the presence of LIF.

3. Figure 2A- going back to the proliferation results, how would the OCR profile appear for Stat3+/+2i and Stat3-/-2i naïve mPSCs? These should appear similar to Stat3-/-2i + LIF and not Stat3+/+2i + LIF naïve mPSCs if the extent of respiration has a role in the rate of proliferation, a main thesis of the current study.

As requested, we compared the OCR profile of Stat3 -/- cells in either 2i or 2i+LIF and Stat3 +/+ cells in 2i appear very similar (Figure S3D).

4. Figure 2D- The number of copies of mtDNA can vary with the mitochondrial protein biomass inside cells, so it is not clear whether a semi-quantitative measurement of mtDNA copy number means much as a surrogate for the number of mitochondria (or the overall network biomass) or for mitochondrial biogenesis.

Although we agree with the referee that the number of mtDNA copies could vary with the mitochondrial biomass, our results show that the number of mtDNA copies is not affected by LIF treatment, suggesting that the changes we observed in mitochondrial transcripts are due to an increased rate of transcription. However, the referee's comment prompted us to examine the expression levels of two mitochondrial basal components, TOM20 and TIMM23, as a proxy for mitochondrial biomass. We observed that LIF treatment had no effect on the level of either protein (Figure 3E). This suggests that LIF does not substantially alter mitochondrial biogenesis.

5. Page 5 - rationale for focusing on complex I abundance- this makes sense if direct Stat3 regulation of the mtDNA transcriptome is the underlying cause (see point 2 above), but one could also argue that the ultimate electron donating complex, IV, is just as compelling.

Indeed we focused on Complex I because it has the largest number of mitochondrially encoded subunits directly affected by LIF/Stat3 (Figure 1D).

Furthermore, complex I is the least well defined and understood of all the ETC complexes in the respiratory chain and can have variable components dependent upon cell type and state. Also, since Stat3 has already been shown to interact in one experimental setting with complex I proteins (Wegryzn, 2009), this and not affects on mtDNA transcription could be the source of reduced complex I levels or supercomplex formation (i.e. enhanced instability and/or turnover). How does complex II abundance appear, since it is entirely nucleus encoded?

As suggested by the referee we measured the abundance of the complex II by western blot and observed a mild reduction in absence of either LIF or in the absence of Stat3, but such differences were not statistically significant (Figure S3F). This is consistent with our results because Complex II components are poorly represented among Stat3 induced genes

6. Figure 2F- The blue native gels that form the basis for supercomplex comparisons should be shown in the supplement.

We moved the panel to the supplement as requested (Figure S3E).

7. Text page 5 and Figure S3A- the claim is that rotenone decreased OCR by >85% but the graph and legend states a 70% reduction- need to make this consistent.

We erroneously referred to the Antimycin A effect (figure S4D), which is indeed >85%. We have corrected the text.

8. Figure 3- why is efficient respiration important for naïve mPSC proliferation? Is it because ATP is generated (FCCP results in Figures 2A, 2B suggest coupling for the ETC) or because ROS is suppressed (complex inhibitors like rotenone increase ROS)? Or because respiration is linked through ETC complex II to the TCA cycle?

This is an interesting issue to which we currently have no definitive answer.

We measured the levels of ROS upon treatment with three different respiratory chain inhibitors, as also suggested by referee #1. None of the three inhibitors increased ROS production at the concentrations used for our biological assays (Figure S5A-D).

We measured ATP levels via a chemoluminescence assay based on firefly luciferase (ATPlite - PerkinElmer). These results are inconclusive. In future studies it will be valuable to use alternative methods to measure ATP levels but this extends beyond the scope of the current manuscript. We

therefore no longer mention “efficient energy production” in the text, but simply state that LIF/Stat3 affect ETC activity.

We agree that the differential ETC activity could influence the TCA cycle or other metabolic pathways such as glycolysis or beta-oxidation. We believe that a comprehensive study of the effects of LIF/Stat3 activity on the metabolome of ES cells would be required to understand how ETC activity is linked to proliferation. Such studies will be interesting to pursue in the future. We have added a final paragraph on this point to the Discussion

Also, rotenone and antimycin target different ETC complexes and have different effects and toxicities, so the comment on bottom of page 5 that (paraphrasing) antimycin was used to rule out off target rotenone effects (assume off-ETC effects is what the authors' mean here) does not make much sense even with both having on-target effects only for the ETC.

We apologize for the lack of clarity if this sentence; in fact we did not intend “off-ETC effects”. All inhibitors used for cellular assays suffer from potential off-target effects. For example, rotenone could inhibit some unknown kinase that could affect proliferation. However, by now showing that three distinct inhibitors of the ETC complexes display the same effects on ES cell proliferation we can exclude unknown off-target effects (Figure 4B, 4F, and S4F). Furthermore, as suggested by referee #1 we also knocked down a Complex I subunit and found reduced OCR levels and proliferation (Figure 4C-E and S4C).

9. Figure 3D- what are the long term consequences of rotenone on cell viability? In the 48h assays at the doses used it looked OK, but what about at P3 or P4? These drugs do not wash out and they may accumulate over time and by P4 there are ~20 to 100-fold less cells in the rotenone treated wells than the 2i+LIF only wells- is cell death a part of this slow down?

We tested the effects of prolonged rotenone inhibition on cell viability by PI/Annexin-V staining and found no significant effects after 4 passages (Figure S5E). We do not find this result surprising because we titrated all the inhibitors in order to find the minimal concentration affecting respiration and proliferation without effecting survival. The concentrations we used in our assays are therefore from 10- to 100-fold lower than those commonly used in cellular biochemistry. Moreover, murine ES cells proliferate very rapidly, therefore the inhibitors may not accumulate significantly because of constant synthesis of new ETC complexes.

10. Text comment page 6- the authors' claim transfected Stat3 is “almost at endogenous levels” for Figure 4B, in which their levels are ~2-fold more than endogenous, whereas the claim is that Stat3 +LIF “significantly” elevated mtRNA levels in Figure 1D when these levels are for many genes ~2 fold over uninduced levels. It seems to this reviewer that 2-fold protein overexpression could have significant unintended effects on cell physiology when compared to 2-fold changes in steady-state RNA expression levels.

We have amended the text to be consistent and more accurate throughout.

11. Figure 4E- the claim is that the Stat3 rescued clones showed reduced sensitivity to rotenone, but the ratio of reduction at 50nM and 100nM rotenone for the Stat3^{-/-} control and especially Stat3.A look similar, meaning a similar effect at a different baseline for these two lines. The same is seen for Figure 4J in which the slope between 50nM and 100nM rotenone is the same whether the cells are Stat3^{-/-} or the 3 MLS-Stat3 rescue clones, indicating that rotenone is having the same dose-dependent activity just from different starting points.

We now amended the text to clarify this point. We agree that the ratio of reduction is similar among the different cell lines, so the word “sensitivity” is inappropriate.

12. Figure 4F- Most mitochondria targeted proteins are translocated coupled to their translation on or near the mitochondria and they are translocated in a chaperone-bound unfolded configuration to

get across the outer (and inner) mitochondrial membranes. If MLS-Stat3 is shipped into the mitochondria, and endogenous Stat3 is usually phosphorylated by JAKs linked to proline-rich regions of intracellular domains of cytoplasmic membrane receptors, then how does MLS-Stat3 become phosphorylated? Is JAK also in/on mitochondria? Does this mean that some MLS-Stat3 is not unfolded and localized to mitochondria?

The results showing phosphorylation of a minor fraction of MLS-Stat3 are intriguing but not critical to our main conclusions. It could be that JAK or some other kinase phosphorylates Stat3 within mitochondria, or that some MLS-Stat3 escapes translation-coupled translocation to mitochondria and becomes phosphorylated via association with the cell surface receptor (upon which it may or may not be able to translocate into mitochondria).

13. The claim throughout is that Stat3 directly supports transcription of the mtDNA. However, this is not directly shown and all assays used to assess mtDNA transcripts do not consider that Stat3 could enhance mitochondrial transcript stability, which is a very complex and active area of investigation (e.g. how are mtDNA transcripts turned over?). Again, the only published reports on Stat3 in mitochondria, albeit in different cellular contexts, do not show a role in mtDNA transcription for enhanced mitochondrial function, even in all of the papers the authors' cite to support or imply this role.

Indeed mitochondrial transcripts could be regulated at the level of both transcription and stability. Our results (see point 2) provide strong evidence for the former but do not rule out that the latter may also play a role. We have amended the Discussion section to include this possibility.

14. Figure 4I- what is the MLS-Stat3 + 2i naïve mPSC proliferation rate for the 3 clones shown without LIF?

We tested the proliferation of MLS-Stat3 clones with and without LIF and found that only in the presence of LIF do they proliferate at a rate similar to Stat3^{+/+} cells (Figure S7A). In the absence of LIF the MLS-Stat3 clones proliferated as Stat3^{-/-} cells. This indicates that LIF is required for the biological activity of MLS-Stat3.

We attempted to dissect how LIF regulates MLS-Stat3. We first hypothesised that LIF might affect the phosphorylation status of MLS-Stat3, in particular at the Ser727 residue, that was reported to be required for its mitochondrial activity (Wegrzyn et al, 2009). However we did not detect any phosphorylation of MLS-Stat3 at that residue, probably because of the presence of the Mek inhibitor (one of the two inhibitors used in 2i media), which has been shown to abolish Ser727 phosphorylation (Huang et al., 2014 – Stem Cells). We then considered that the level of Stat3 protein might be affected by LIF and indeed observed that active LIF signalling leads to increased Stat3 protein, both for endogenous Stat3 and exogenous MLS-Stat3 protein (Figure S7B). We therefore suggest that the positive effect of LIF on proliferation of MLS-Stat3 clones could be due to an increase in the protein levels of MLS-Stat3. How this is achieved remains unclear but we note that LIF stimulates PI3K signalling and this will be active in Stat3 null ES cells in 2i.

15. The claim is that LIF increases the efficiency of energy production to support rapid naïve mPSC proliferation but there are no measurements to compare ATP levels in any of the conditions tested that manipulate LIF, Stat3, or the ETC. See point 8 above also....

As stated in point 8, we were not able to detect a change in ATP levels after treatment with ETC inhibitors, but we do not find such results conclusive the sensitivity of the assay may be limiting. As discussed above we have now removed the exclusive emphasis on bioenergetics and present other possibilities for metabolic influences on ES cell proliferation.

Based on our results and referee #2 comments, now we do not mention “efficient energy production” in text, but simply state that LIF/Stat3 affect ETC activity, for which we provided solid evidence.

2nd Editorial Decision

12 January 2016

Thank you for the submission of your revised manuscript to The EMBO Journal. As you will see below, your article was sent back to two of the original referees, who now consider that you have properly dealt with the main concerns originally raised in the review process, and therefore I am writing with an 'accept in principle' decision. This means that I will be happy to formally accept your manuscript for publication once a few more minor issues have been addressed.

As I said, referees now believe that all major concerns have been addressed and your manuscript is almost ready for publication (see below). A few relatively minor points are still pending according to referee #1. In essence, these can be summarized in two actions: please, tone down the conclusions regarding transcriptional regulation in the mitochondrion by STAT3, and provide evidence supporting your lack of toxicity statements concerning the inhibitors used by titration experiments as suggested by the referee.

I would also like to point out to a very minor issue with figures 2, 4, 5, 6, S2, S6 and S8: micrographs, including confocal or immunofluorescence images, must include scale bars for clarity, and their size must be stated in the corresponding figure legend.

Thank you very much for your patience. I am looking forward to seeing the final version of your manuscript. Congratulations in advance for a successful publication.

REFEREE REPORTS

Referee #1:

The revised manuscript by Carbognin and colleagues supports a key role for the maintenance of mitochondrial respiratory activity supported by mitochondria-dependent STAT3 mechanisms on ESC cell survival and naive pluripotency. The additional experiments with respiratory inhibitors and knockdown of a complex I subunit are appreciated and helpful. The lack of observed alterations in ROS production are most likely due to the relative insensitivity of the assay method, though this is not a major point. The data of STAT3 mediated upregulation of mitochondrial transcripts is intriguing, but continues to have multiple potential mechanisms. Nucleoid data is of interest, but remains indirect. The conclusions that mitochondrial STAT3 directly regulates mitochondrial transcription are not adequately supported and should be removed. The data clearly support that LIF-driven via STAT3 or MLS-STAT3 driven support of respiration are critical to stem cell homeostasis and proliferation are clearly supported and are a major strength of the paper. As a supplemental data, the inhibitor titrations of the pharmacologic inhibitors should be shown to support the technical comments made regarding relative lack of toxicity.

Referee #2:

The authors have adequately addressed the issues left unresolved in the first submission with this improved revised submission. The quality and consistency of the data is also improved.

2nd Revision - authors' response

14 January 2016

Response to reviewers

We thank the referees for carefully considering the revised version of our manuscript. Below is a point-by-point response.

Referee #1:

The revised manuscript by Carbognin and colleagues supports a key role for the maintenance of mitochondrial respiratory activity supported by mitochondria-dependent STAT3 mechanisms on ESC cell survival and naive pluripotency. The additional experiments with respiratory inhibitors and knockdown of a complex I subunit are appreciated and helpful. The lack of observed alterations in ROS production are most likely due to the relative insensitivity of the assay method, though this is not a major point. The data of STAT3 mediated upregulation of mitochondrial transcripts is intriguing, but continues to have multiple potential mechanisms. Nucleoid data is of interest, but remains indirect. The conclusions that mitochondrial STAT3 directly regulates mitochondrial transcription are not adequately supported and should be removed. The data clearly support that LIF-driven via STAT3 or MLS-STAT3 driven support of respiration are critical to stem cell homeostasis and proliferation are clearly supported and are a major strength of the paper. As a supplemental data, the inhibitor titrations of the pharmacologic inhibitors should be shown to support the technical comments made regarding relative lack of toxicity.

The assay we performed to measure ROS levels is widely used, and we included appropriate positive controls (e.g. high doses of Rotenone) to make sure we were able to detect changes in ROS production. However, we agree that other assays might be more sensitive and could potentially detect minor changes in ROS levels. We amended the text accordingly.

We changed our conclusions on the effect of Stat3 on mitochondrial transcripts by deleting the word "directly" from the title and by rewording several sentences throughout the text. Now we refer to "altered expression levels" rather than "enhanced transcription" in order to not exclude other possible mechanisms (such as RNA stability or turnover). Relevant changes are now highlighted in the Manuscript file.

We should also point out that in the Discussion of the revised version of the manuscript we already stated that:

Our results do not exclude involvement of other mechanisms, such as direct interaction between Stat3 and Complex I, or regulation of mitochondrial transcript stability by Stat3, which could synergize with the transcriptional effects we observe.

In sum, now we claim, as the referee suggested, that our results are compatible with multiple mechanisms, such as transcriptional regulation or regulation of RNA stability/turnover.

We now show the titration experiments performed with the inhibitors of the respiratory chain (Appendix Fig. S4A,E and S5A), as requested.

For clarity we should mention that we already provided in the revised version of the manuscript results indicating lack of toxicity at the concentrations used in our assays (Figure 4A, Appendix Fig S4G, S6E, S10F and S10G) by measuring the percentage of PI/Annexin-V positive cells upon treatment with the different inhibitors.

Referee #2:

The authors have adequately addressed the issues left unresolved in the first submission with this improved revised submission. The quality and consistency of the data is also improved.

We thank Referee #2 for appreciating and acknowledging our efforts to improve the manuscript according to his/her suggestions.

3rd Editorial Decision

15 January 2016

I am pleased to inform you that your manuscript has been accepted for publication in the EMBO Journal.